# Accelerated epigenetic aging in Huntington's disease involves polycomb repressive complex 1

Baptiste Brulé [1,2,3], Rafael Alcalá-Vida [1,2,3,8], Noémie Penaud [1,2,3], Jil Scuto [1,2,3], Coline Mounier[1,2,3], Jonathan Seguin [1,2,3], Sina Vincent Khodaverdian[3,4], Brigitte Cosquer[1,2,3], Etienne Birmelé[3,4], Stéphanie Le Gras [3,5,6,7], Charles Decraene[1,2,3], Anne-Laurence Boutillier [1,2,3] & Karine Merienne [1,2,3] ✉

Loss of epigenetic information during physiological aging compromises cellular identity, leading to de-repression of developmental genes. Here, we assessed the epigenomic landscape of vulnerable neurons in two reference mouse models of Huntington neurodegenerative disease (HD), using cell-type-specific multi-omics, including temporal analysis at three disease stages via FANS-CUT&Tag. We show accelerated de-repression of developmental genes in HD striatal neurons, involving histone re-acetylation and depletion of H2AK119 ubiquitination and H3K27 trimethylation marks, which are catalyzed by polycomb repressive complexes 1 and 2 (PRC1 and PRC2), respectively. We further identify a PRC1-dependent subcluster of bivalent developmental transcription factors that is re-activated in HD striatal neurons. This mechanism likely involves progressive paralog switching between PRC1-CBX genes, which promotes the upregulation of normally low-expressed PRC1-CBX2/4/8 isoforms in striatal neurons, alongside the down-regulation of predominant PRC1-CBX isoforms in these cells (e.g., CBX6/7). Collectively, our data provide evidence for PRC1-dependent accelerated epigenetic aging in HD vulnerable neurons.

Huntington's disease (HD) is a genetic, progressive neurodegenerative disease, which manifests with characteristic motor, cognitive, and psychiatric symptoms that arise generally at midlife and progress during 10 to 15 years, with subtle signs that can appear decades earlier. The disease is caused by an abnormal expansion of CAG trinucleotide repeats in the Huntingtin (*HTT*) gene, which produces mutant HTT with polyglutamine expansion (polyQ)[1,2]. Specific neurons, particularly direct and indirect spiny projection neurons (dSPN and iSPN,

respectively) in the striatum, representing >95% of striatal neurons, are highly vulnerable to the HD mutation[3,4]. Molecular underpinning of striatal neuron vulnerability in HD remains elusive.

Epigenetic alteration is a key hallmark of cellular aging[5,6]. Recent study shows that cellular aging is caused by loss of cellular identity resulting from erosion of epigenetic information, so-called exdifferentiation[7]. Histone modifications, including the active mark H3K27 acetylation (H3K27ac) and the repressive mark H3K27

[1]Laboratoire de Neurosciences Cognitives et Adaptatives (LNCA), Strasbourg, France. [2]Centre National de la Recherche Scientifique (CNRS, UMR 7364), Strasbourg, France. [3]University of Strasbourg, Strasbourg, France. [4]IRMA, Strasbourg, France. [5]Institut de Genetique et de Biologie Moleculaire et Cellulaire, Strasbourg, France. [6]CNRS UMR7104, Strasbourg, France. [7]INSERM U1258, Strasbourg, France. [8]Present address: Instituto de Neurociencias (Universidad Miguel Hernández - Consejo Superior de Investigaciones Científicas). Av. Santiago Ramón y Cajal s/n. Sant Joan d'Alacant, Alicante, Spain. ✉e-mail: karine.merienne@unistra.fr

trimethylation (H3K27me3), were used as proxy to assess exdifferentiation, which is characterized by de-repression of non-lineage-specific genes, particularly developmental genes, and repression of cellular identity genes, both mechanisms contributing to cellular identity loss[7].

Epigenomic analyses in bulk HD striatal tissue show early loss of H3K27ac at striatal super-enhancers -broad enhancers regulating cellular identity genes, which results in transcriptional downregulation of their targets, striatal identity genes[8–12]. However, due to lack of cellular and temporal resolutions of earlier analyses, it was unclear whether HD vulnerable neurons undergo accelerated epigenetic aging, including accelerated de-repression of developmental genes.

Here, we investigated HD striatal epigenetic landscape using fluorescence-activated nuclei sorting (FANS) coupled to ChIPseq or CUT&Tag. We profiled neuronal and non-neuronal striatal cells in two reference HD mouse models, R6/1 transgenic and Q140 knockin (KI) mice, showing distinct disease progressions. Additionally, we performed temporal analysis at 3 ages, corresponding to prodromal, early and advanced pathological stages in HD Q140 mice. We targeted H3K27me3, histone acetylation marks (e.g., H3K27ac, H3K9ac, H3K18ac) as well as H2AK119ub, since our analyses highlighted altered regulation of PRC1 subunits in HD striatal neurons. Our data show that the HD mutation accelerates loss of epigenetic information associated with cellular aging in striatal neurons of HD mice. Particularly, epigenetic de-repression of developmental genes is dramatically accelerated and our data indicate that the mechanism involves PRC1 through paralog switching among PRC1-CBX genes. Together, our data support the view that PRC1-mediated accelerated epigenetic aging is a key mechanism underlying striatal neuron vulnerability in HD.

## Results

### Developmental genes are de-repressed in HD striatal neurons

To investigate the epigenetic status of developmental genes in HD striatal cells, we generated ChIP-seq or CUT&Tag data on NeuN+ and NeuN- nuclei prepared from striatal tissue of 15 to 20 week old symptomatic HD R6/1 transgenic and control mice (Fig. 1a, b and Supplementary Fig. 1a–c). We first focused on cellular identity-associated marks (i.e., H3K27ac and H3K27me3), which were recently used to assess epigenetic aging[7]. Consistent with earlier bulk H3K27ac ChIPseq analyses[8–12], H3K27ac signals in R6/1 striatal neurons was reduced at neuronal-specific enhancers and SPN identity genes regulated by super-enhancers (e.g., *Drd2, Drd1*), which associated with increased H3K27me3 levels (Fig. 1b, c and Supplementary Fig. 1c, d). The effect appeared specific to neurons since non-neuronal-specific enhancers did not show loss of H3K27ac and gain of H3K27me3 in R6/1 vs WT NeuN- samples (Supplementary Fig. 1d). The numbers of neurons, including SPN (CTIP2+), were not different between R6/1 and WT mice, indicating that histone mark changes resulted neither from altered cell-type proportion or neuronal loss (Supplementary Fig. 1e).

Further, differential analysis showed thousands of H3K27ac differentially enriched regions (DER) in R6/1 vs WT neurons (NeuN+), including both depleted and enriched regions, whereas few DER were identified (<100) between R6/1 and WT non-neuronal cells (NeuN-) (Fig. 1d and Supplementary Table). H3K27me3 ChIPseq data revealed similar pattern, with hundreds of DER in R6/1 vs WT neurons, and almost no change in NeuN- cells (Fig. 1d and Supplementary Table). These data indicate that epigenetic dysregulation is dramatic in neurons and minimal in non-neuronal cells in HD mouse striatum. However, due to heterogeneity of NeuN- nuclei, we cannot fully rule out that specific non-neuronal cells (e.g., glial cells) undergo substantial epigenetic changes. Integration with transcriptomic data generated in dSPN and iSPN of R6/2 line[11], a sister model to R6/1 mouse[13], showed that downregulated genes in dSPN and iSPN expressing the HD gene were depleted in H3K27ac and enriched in H3K27me3, while upregulated genes were enriched in H3K27ac and depleted in H3K27me3

(Fig. 1e). Thus, as expected, H3K27ac and RNA changes in HD mouse striatal neurons were positively correlated, whereas H3K27me3 and RNA changes were anti-correlated.

We then performed functional enrichment analysis, which revealed that H3K27me3-depleted genes in R6/1 vs WT striatal neurons were particularly enriched in biological processes (GO BP) related to neuronal development and differentiation, including "generation of neurons", "neuron differentiation" and "nervous system development" (Fig. 2a). Analyses using transcription factor (ChEA) and HD-related signature (HDsigDB) databases further indicated that H3K27me3-depleted genes were regulated by PRC1 (e.g., RNF2, RING1B) and PRC2 (e.g., SUZ12, JARID2) and presented bivalent signature (i.e., enriched in both H3K27me3 and H3K4me3) (Fig. 2b and Supplementary Fig. 2a). H3K27ac-increased genes in HD neurons also showed developmental signature (Supplementary Fig. 2b). Further, 42% of H3K27me3-depleted genes in HD neurons also gained H3K27ac (Fig. 2c), identifying subset of 125 euchromatinized genes linked to neurodevelopment (Fig. 2d and Supplementary Fig. 2c). Those 125 genes were significantly upregulated in dSPN and iSPN of HD mice (Fig. 2e).

We further defined histone acetylation landscape of striatal neurons in R6/1 mice, generating H3K9ac and H3K18ac FANS-CUT&Tag data (Fig. 1a, Supplementary Fig. 2d, e and Supplementary Table). The CUT&Tag protocol[14] was optimized to enable sample processing after FANS (Methods). The data showed that H3K9ac and H3K18ac, in addition to H3K27ac, were increased at the 125 neurodevelopment-related genes in HD mice, suggesting general histone re-acetylation and decreased H3K27me3 contributes to their transcriptional de-repression (Fig. 2f). Moreover, functional enrichment analysis supported strong overlap between development-related genes showing increased H3K9ac, H3K18ac and H3K27ac in R6/1 vs WT striatal neurons (Fig. 2g). Epigenetic de-repression of developmental genes in R6/1 mouse striatal neurons is exemplified with the developmental transcription factor *Onecut1*, its upstream regulator *Pax6*, and *Runx2*, forming strong interaction network[15] (Fig. 2e, h, i and Supplementary Fig. 2f). Supporting the results, *Onecut1*, *Pax6* and *Runx2* were transcriptionally increased in the striatum of HD mice, particularly in iSPN (Fig. 2j and Supplementary Fig. 2g). Together, our data indicate that the HD gene leads to heterochromatinization of identity genes and euchromatinization of developmental genes in vulnerable neurons, resembling aging-associated loss of epigenetic information.

### Bivalent promoters are de-repressed in HD striatal neurons

Above analyses predict that genes susceptible to euchromatinization in HD striatal neurons are regulated by bivalent promoters. To test the hypothesis, we carried out kmeans clustering analysis using striatal H3K4me3 and H3K27me3 ChIPseq data, which allowed the identification of bivalent cluster strongly enriched in PRC1/PRC2-regulated developmental genes, including *Onecut1*, *Pax6* and *Runx2* (Fig. 3a, Supplementary Fig. 3a–c and Supplementary Table). The result is consistent with studies showing that PRC2 and PRC1 frequently overlap at promoters of developmental genes that need to be repressed in mature differentiated cells[16]. Further, metaprofile analysis focused on bivalent gene promoters showed selective decrease in H3K27me3 and increase in histone acetylation in R6/1 vs WT striatal neurons, supporting specific euchromatinization of bivalent genes in HD vulnerable neurons (Fig. 3b, c and Supplementary Fig. 3d). Integration with transcriptomic data[11,17] strengthened the conclusion, showing transcriptional de-repression of a majority of bivalent genes, particularly in iSPN, in both HD mice and HD patients (Fig. 3d–f).

### PRC1-CBX genes undergo paralog switch in striatal neurons of HD mice

Altered regulation of polycomb group (PcG) proteins, which are subunits of PRC1 and PRC2, might underlie euchromatinization of bivalent genes in HD striatal neurons. EZH1, the predominant enzymatic

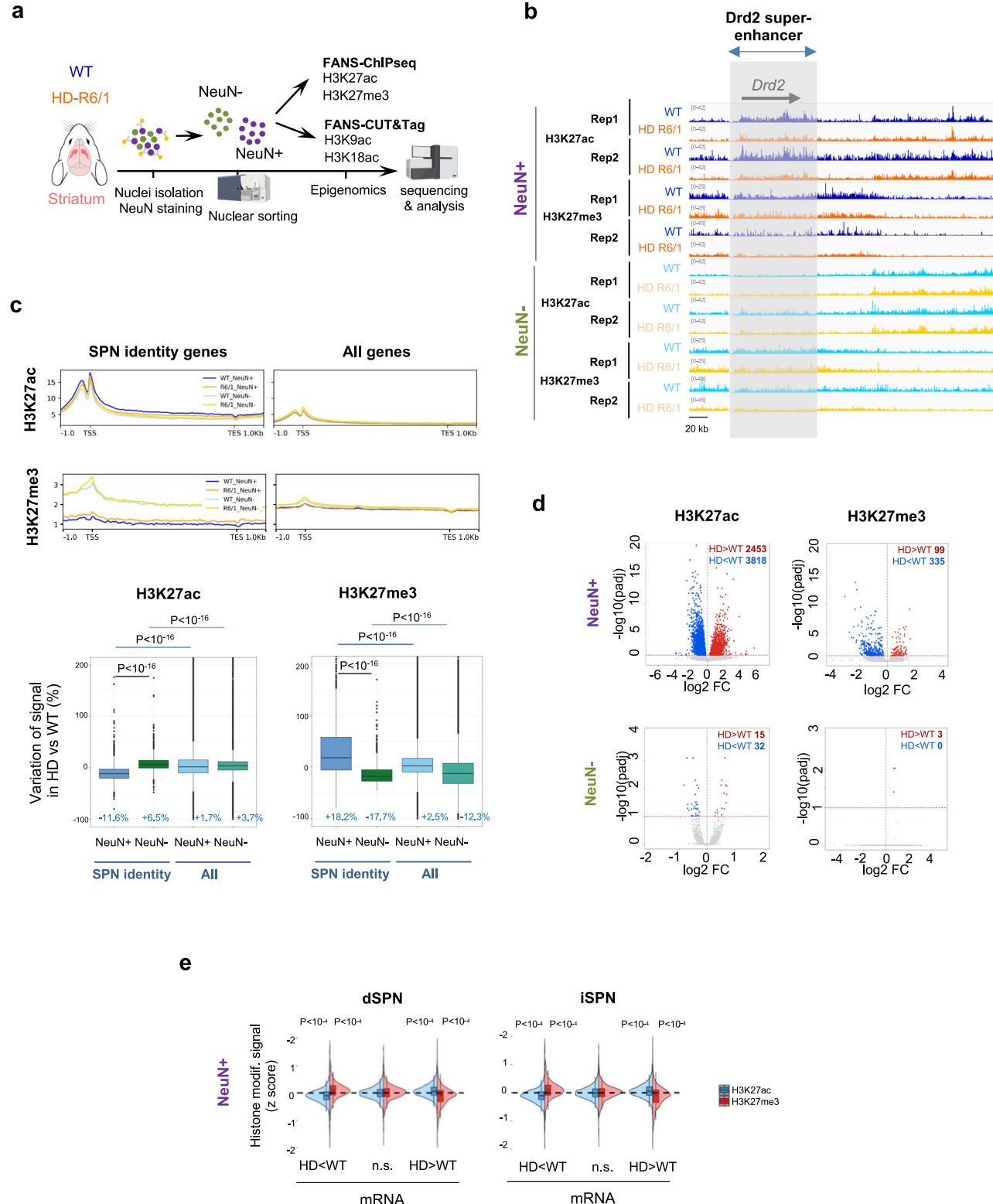

subunit catalyzing H3K27me3 in striatal neurons, was slightly reduced in HD mouse striatum, both at protein and mRNA levels (Supplementary Fig. 4a, b), suggesting impaired PRC2 activity might contribute to bivalent gene de-repression in HD vulnerable neurons. However, the other PRC2 subunits were not consistently changed (Supplementary Fig. 4a, b). Additionally, although H3K27me3 is strongly correlated with low transcriptional activity, this PRC2-mediated histone modification alone is not sufficient for gene repression, in contrast to PRC1[16]. Interestingly, we found that the top region showing increased

chromatin relaxation in R6/1 vs WT striatal neurons, with most significant adjusted *p*-values for both depletion in H3K27me3 and enrichment in H3K27ac, was an enhancer located in the genomic region that contains PRC1 *Cbx2/4/8* paralogous genes, between *Cbx4* and *Cbx8* (Fig. 4a). Additional significantly de-repressed region was an enhancer located upstream of *Cbx4* promoter (Fig. 4a). *Cbx2, Cbx4* and *Cbx8* are substantially expressed in differentiating cells, including neural progenitor cells (NPC), and become downregulated in differentiated cells[18]. Consistently, *Cbx2, Cbx4* and *Cbx8* expression was low

**Fig. 1 | Cellular identity-associated histone marks are specifically altered in striatal neurons of HD mice. a** Scheme illustrating FANS-ChIPseq and FANS-CUT&Tag experiments conducted on 15–20 week-old R6/1 and WT mice (Created in BioRender. Boutillier, A. (2025) https://BioRender.com/e61g067). H3K27ac and H3K27me3 FANS-ChIPseq, $N = 2$ biological replicates; H3K9ac FANS-CUT&Tag, $N = 5$ biological replicates (WT) and $N = 4$ biological replicates (R6/1); H3K18ac FANS-CUT&Tag, $N = 2$ biological replicates. **b** IGV genome browser capture showing H3K27ac and H3K27me3 signals in R6/1 and WT NeuN+ and NeuN- samples at super-enhancer-regulated SPN identity gene locus (e.g., *Drd2*). The grey box highlights *Drd2* super-enhancer. Rep1, biological replicate 1; rep2, biological replicate 2. **c** Top, metaprofiles showing H3K27ac and H3K27me3 mean signals in WT and R6/1 NeuN+ and NeuN- nuclei, along SPN identity genes (left) and all genes (right) in mouse striatum. SPN identity genes correspond to module M2 in WGCNA study by ref. 10. Bottom, boxplots showing H3K27ac and H3K27me3 signal variations, expressed as percentages, in R6/1 *vs* WT at SPN identity genes and all genes. Statistical analysis was performed using Kruskal–Wallis test and Bonferroni correction for multiple testing. Boxplots show median, first quartile (Q1), third quartile (Q3)

and range (min, Q1-1.5*(Q3-Q1); max, Q3+1.5*(Q3-Q1). Median values are indicated in blue. H3K27ac, $N = 2$ biological replicates in each group; H3K27me3, $N = 2$ biological replicates in each group **d** Volcano plots showing H3K27ac (left) and H3K27me3 (right) differentially enriched regions in R6/1 *vs* WT for NeuN+ (top) and NeuN- (bottom) samples. Decreased and increased regions in R6/1 vs WT are represented in blue and red, respectively (DESeq2 method, adj. *P*-val < 0.1 using the Benjamini–Hochberg method for multiple testing correction). **e** R6/1 NeuN+ H3K27ac (blue) and H3K27me3 (red) ChIPseq signals in downregulated (HD<WT), Non-significantly changed (n.s.) or upregulated (HD>WT) genes in dSPN (left) and iSPN (right) of HD R6/2 mice. Integration with RNAseq data by ref. 11 (dSPN, $N = 7$ biological replicates in each group; iPSN, $N = 4$ biological replicates in each group). Boxplots show median, first quartile (Q1), third quartile (Q3) and range (min, Q1-1.5*(Q3-Q1); max, Q3+1.5*(Q3-Q1). Statistical analysis was performed using Mann–Whitney test (two-sided). H3K27ac, $N = 2$ biological replicates in each group; H3K27me3, $N = 2$ biological replicates in each group. Source data are provided as a source data file.

in mature striatal neurons (Fig. 4b and Supplementary Fig. 4c). Euchromatinization of *Cbx2/4/8* enhancers in HD mouse vulnerable neurons correlated with transcriptional de-repression of *Cbx2*, *Cbx4* and *Cbx8* (Fig. 4b, c and Supplementary Fig. 4c). Supporting those results, protein analyses showed that CBX4 and CBX8 proteins were increased in the striatum of HD mice (Fig. 4d, e). Further, IHF experiment showed that CBX8 was increased in striatal neurons of R6/1 vs WT mice (Fig. 4f). In contrast, we found that *Cbx6* and *Cbx7*, particularly *Cbx6*, are highly expressed in mature SPN, indicating they are predominant PRC1 CBX proteins in these neurons (Fig. 4b and Supplementary Fig. 4c). Strikingly, *Cbx6/7* are located in distinct genomic region compared to *Cbx2/4/8*, suggesting different regulatory mechanisms. In support to this hypothesis, CBX6 was reduced in the striatum of HD mice, essentially due to downregulation in SPN, and CBX7 showed similar tendency (Fig. 4b–e and Supplementary Fig. 4c). Furthermore, *Cbx6* was depleted in H3K27ac in R6/1 striatal neurons (Supplementary Fig. 4d). Together, this indicates that PRC1-CBX genes undergo paralog switch in HD SPN, promoting stoichiometry observed in immature neurons[18]. Finally, temporal transcriptomic data generated on striatal tissue of HD knockin (KI) mice, including Q140 and Q175 lines[10], and proteomic data generated on the striatum of the HD-R6/2 line both showed age-dependent increase of *Cbx4* and *Cbx8* and reduction of *Cbx6* and *Cbx7* in HD mice, suggesting progressive PRC1-CBX paralog switch (Fig. 4e and Supplementary Fig. 4e). The mechanism appeared attenuated in the cortex, a tissue that is less affected than the striatum (Supplementary Fig. 4e). Thus, PRC1-CBX paralog switch might be implicated in developmental gene de-repression in HD striatal neurons.

## H2AK119ub is depleted at subcluster of bivalent promoters in the striatum of HD mice

PRC1 mediates chromatin repression catalyzing H2AK119ub[19]. We first generated H2AK119ub ChIPseq data on bulk striatal tissue of R6/1 and WT mice, which showed that H2AK119ub was dysregulated in the striatum of HD R6/1 mice (Fig. 5a and Supplementary Fig. 5a, b and Supplementary Table). Remarkably, H2AK119ub-depleted regions were greater than H2AK119ub-enriched regions in R6/1 vs WT mice (i.e., 2274 vs 636, Supplementary Fig. 5a). H2AK119ub was increased at genes related to TGFβ signaling and developmental processes associated with non-neuronal lineage specification in R6/1 striatal tissue (Supplementary Fig. 5c). In contrast, genes associated with depleted H2AK119ub in R6/1 striatal tissue were enriched in neurodevelopmental genes (Supplementary Fig. 5c). To specify the signature of genes depleted in H2AK119ub in R6/1 striatum, we investigated H2AK119ub at bivalent promoters. Kmeans clustering analysis identified 2 clusters, clusters 1 and 2, characterized by high and low H2AK119ub levels, respectively (Fig. 5b and Supplementary Table).

Cluster 1 was strongly enriched in neurodevelopmental transcription factors (e.g., "Neuron differentiation", "Generation of neurons", "Regulation of transcription by RNA PolII"), whereas cluster 2 was enriched in GO terms related to neuronal plasticity (e.g., "synapse pruning") (Fig. 5c and Supplementary Fig. 5d). H2AK119ub was specifically reduced at cluster 1 promoters in R6/1 vs WT striatal samples (Fig. 5b). In contrast, H3K27me3 depletion in R6/1 vs WT striatal neurons was similar in both clusters (Fig. 5d). Additionally, H3K27ac was similarly increased in both clusters in R6/1 striatal neurons (Fig. 5d). Kmeans clustering of bivalent genes using H3K27me3 data, instead of H2AK119ub data, supported the results, showing specific decrease of H2AK119ub in gene cluster normally strongly repressed (Supplementary Fig. 5f). Finally, to better compare H2AK119ub and H3K27me3 changes in R6/1 striatal neurons, we generated H2AK119ub and H3K27me3 FANS-CUT&Tag data using same NeuN+ nuclei (Supplementary Fig. 5g, h). Data analysis confirmed that H2AK119ub, in contrast to H3K27me3, was specifically reduced in cluster 1, in R6/1 vs WT striatal neurons (Supplementary Fig. 5i). Further, integration with transcriptomic data showed that cluster 1 genes showed a tendency to upregulation in iSPN of HD mice (Fig. 5e). Thus, impaired activity of PRC1 in HD vulnerable neurons might drive de-repression of developmental transcription factors, which control neuronal fate.

## De-repression of developmental genes is an aging signature that is accelerated in striatal neurons of HD mice

We then explored whether PRC1-mediated de-repression of developmental transcription factors in HD vulnerable neurons might reflect acceleration of epigenetic aging. To address this question, we generated H3K27ac and H3K27me3 CUT&Tag data using striatal neuronal nuclei of HD Q140 KI and WT mice (Fig. 6a). Q140 KI mouse is a slowly progressing model compared to R6/1 mice, which is particularly suitable for temporal analysis. Epigenomic data were produced at 2-, 6- and 10-month-old, corresponding to prodromal, early symptomatic and more advanced symptomatic stages. Principal component analysis (PCA) analysis showed that H3K27ac and H3K27me3 samples relatively clustered according to genotype and age (Supplementary Fig. 6a). The quality of the data is further illustrated in Fig. 6b, representing snapshot of H3K27ac and H3K27me3 profiles at *Onecut1* locus. Differential analysis showed that DER increased over time, however particularly at 10 months for H3K27ac and from 6 months for H3K27me3 (Supplementary Fig. 6b and Supplementary Table). Consistent with R6/1 ChIPseq data, functional enrichment analyses showed that H3K27ac-enriched regions and H3K27me3-depleted regions in HD KI vs WT mice both displayed significant developmental signatures, particularly at 10 months (Supplementary Fig. 6c, d). Remarkably, H3K27ac and H3K27me3 in HD KI striatal neurons were increased and decreased over time at *Onecut1* locus, respectively (Fig. 6c). Runx2 showed similar

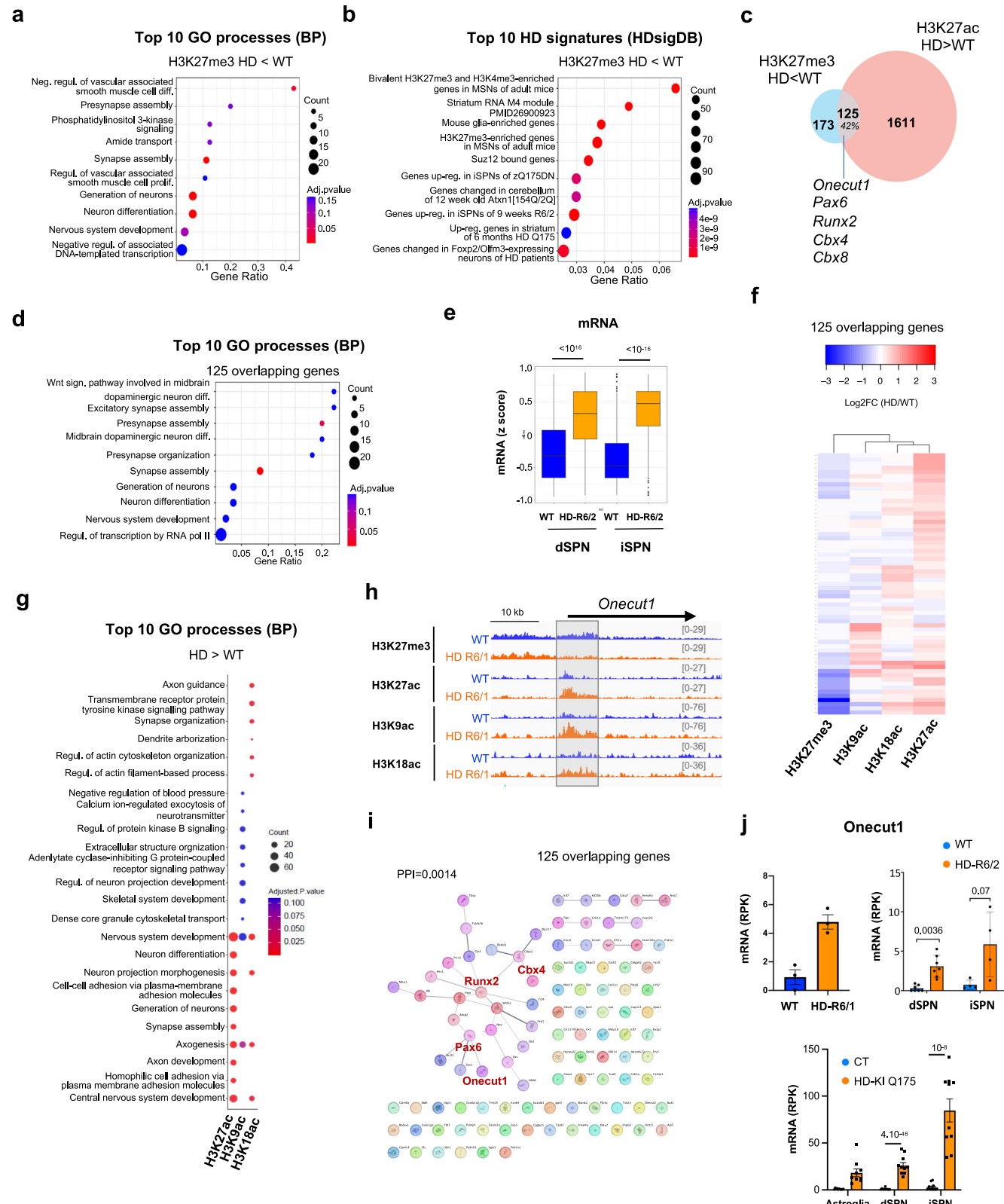

progression (Supplementary Fig. 6e). Computing euchromatin score that combines H3K27ac and H3K27me3 signals showed accelerated euchromatinization of *Onecut1* and Runx2, which correlated with accelerated transcriptional activation (Fig. 6c, d and Supplementary Fig. 6e). Developmental transcription factors in cluster 1 above identified also showed accelerated euchromatinization using euchromatin score (Fig. 6e). In fact, euchromatin score at cluster 1 genes could be

used as a proxy to measure epigenetic age of HD striatal neurons, which was 3 month-older than chronological age of 6-month-old mice (Fig. 6e).

Age-dependent variations of H3K27me3 and H3K27ac in cluster 1 genes suggest that H3K27me3 contributes to epigenetic age in both HD and WT striatal neurons, whereas H3K27ac may have specific role in HD (Fig. 6f). Differential analysis of H3K27me3 and H3K27ac levels in

**Fig. 2 | Developmental genes are epigenetically de-repressed in striatal neurons of HD mice. a** Top 10 gene ontology (GO) processes (biological processes, BP) enriched in H3K27me3-depleted regions in R6/1 vs WT NeuN+ samples. BP terms are shown as a function of gene ratio, gene count and adj. *Pval*. Adj. *Pval* were calculated using the Benjamini−Hochberg method for multiple testing correction **b** Top 10 HD-related signatures (HDsigDB database) enriched in H3K27me3-depleted regions in R6/1 vs WT NeuN+ samples. HDsigDB terms are shown as a function of gene ratio, gene count and adj. *Pval*. Adj. *Pval* were calculated using the Benjamini−Hochberg method for multiple testing correction **c** Overlap between H3K27me3-depleted (HD < WT, blue) and H3K27ac-enriched (HD>WT, red) genes in R6/1 NeuN+ samples. 125 genes, representing 42% of H3K27me3-depleted genes, are enriched in H3K27ac. Overlapping genes include *Onecut1, Pax6, Runx2, Cbx*4 and *Cbx8*. **d** Top 10 gene ontology (GO) processes (biological processes, BP) enriched in 125 overlapping genes. BP terms are shown as a function of gene ratio, gene count and adj. *Pval*. Adj. *Pval* were calculated using the Benjamini−Hochberg method for multiple testing correction **e** Boxplots showing z-scores of expression values for the 125 overlapping genes in dSPN (left) and iSPN (right) in WT and HD R6/2 mice. RNAseq data in ref. [11]. Boxplots show median, first quartile (Q1), third quartile (Q3) and range (min, Q1-1.5*(Q3-Q1); max, Q3+1.5*(Q3-Q1). Statistical analysis was performed using Kruskal−Wallis test and Bonferroni correction for multiple testing. dSPN, *N* = 7 biological replicates in each group; iPSN, *N* = 4 biological replicates in each group. **f** Heatmap representing R6/1 vs WT log2FC in H3K27me3, H3K27ac, H3K9ac and H3K18ac NeuN+ samples at the 125 H3K27me3-depleted and H3K27ac-enriched overlapping genes. **g** Top 10 gene ontology (GO) processes (biological processes, BP) enriched in H3K27ac, H3K9ac and H3K18ac-increased regions in R6/1 vs WT NeuN+ samples. BP terms are shown as a function of gene count and adj. *Pval*. Adj. *Pval* were calculated using the Benjamini−Hochberg method for multiple testing correction **h** IGV genome browser capture showing H3K27me3, H3K27ac, H3K9ac and H3K18ac signals in R6/1 and WT NeuN+ and NeuN- samples at *Onecut1* gene locus. The grey box highlights *Onecut1* promoter. **i** Protein-protein interaction network using STRING on 125 overlapping genes. The major subnetwork, which includes *Onecut1, Pax6, Runx2* and *Cbx4* is highlighted. PPI (protein protein interaction), statistics showing the strength of the network. **j** mRNA levels of *Onecut1* in bulk striatal tissue of R6/1 and WT mice (RNAseq data by ref. [74]; *N* = 3 biological replicates in each group), in dSPN and iSPN of R6/2 and WT mice (RNAseq data by ref. [11]; dSPN, *N* = 7 biological replicates in each group; iPSN, *N* = 4 biological replicates in each group) and in striatal astroglia, dSPN and iSPN of Q175 knockin (HD-KI-Q175) and control (CT) mice (RNAseq data by ref. [11]; Astroglia, *N* = 10 biological replicates in each group; dSPN, *N* = 10 biological replicates in each group; iPSN, *N* = 10 biological replicates in each group). mRNA levels, reads per kilobases (RPK). Mean values +/- sem are shown. Statistics show adj. *Pval*, multiple testing correction was performed using the Benjamini−Hochberg method upon analysis of RNAseq data. Source data are provided as a source data file.

WT samples showed that genes depleted in H3K27me3 and increased in H3K27ac with age, both displayed developmental signatures, including signature related to "Nervous system development" (Supplementary Fig. 6f). Moreover, H3K27me3-depleted genes in HD vs WT and in old vs young samples significantly overlapped (Supplementary Fig. 6g). This was also the case when considering H3K27ac-increased genes (Supplementary Fig. 6g). This indicates that common mechanism contributes to de-repression of developmental genes in normal aging and in HD. Interestingly however, the analysis also suggests greater involvement of H3K27ac in HD and of H3K27me3 in normal aging, since changes in H3K27ac and in H3K27me are predominant in HD and normal aging, respectively (Supplementary Fig. 6g).

Euchromatin score at *Cbx4* and *Cbx8* genes were dramatically increased with age in HD KI mice, and not in WT animals, which might suggest specific role of PRC1-CBX paralog switch in accelerated epigenetic erosion in HD (Fig. 6g, h and Supplementary Fig. 6h, i). Meta-profile analyses further showed that gene promoters in cluster 1 were subject to accelerated euchromatinization in HD mouse striatal neurons (Fig. 6i, j). Thus, euchromatinization rate of developmental transcription factors could be used as an aging index to show acceleration of epigenetic age in HD-vulnerable neurons. The data also suggest specific role for PRC1-CBX proteins in the mechanism.

To further investigate aging-associated chromatin changes in HD KI and WT mouse striatal neurons, we ran co-expression module analysis using euchromatin score, which identified 4 modules of co-regulated genes (Fig. 6k and Supplementary Table). Whereas the 4 modules were age-dependent, M2 and M4 modules also appeared related to genotype (Fig. 6k). Euchromatinization increased over time in M2 genes, however faster in HD KI mice (Fig. 6k). M2 genes, including *Onecut1*, significantly overlapped with genes in cluster 1 and were enriched in bivalent, developmental transcription factors (Fig. 6l and Supplementary Fig. 7a). Additionally, they were transcriptionally upregulated in HD vulnerable neurons (Supplementary Fig. 7b). Thus, co-expression module analysis using euchromatin score confirmed accelerated de-repression of developmental transcription factors in HD vulnerable neurons.

### Epigenetic regulation of stress response is abnormal in HD mouse striatal neurons

M4 genes displayed different pattern than M2 genes. Euchromatin score in M4 progressively decreased with age in HD KI striatal neurons, whereas it peaked at 6 months of age in WT mice (Fig. 6k).

Functionally, M4 genes were related to stress response (Fig. 7a, b and Supplementary Fig. 7c). Specifically, enriched GO BP terms comprised "response to chemokine" and processes related to cell signaling such as "Rac protein signal transduction" (Fig. 7b and Supplementary Fig. 7c). Moreover, top transcriptional regulators of M4 genes were major drivers of stress response, including NR3C1 and STAT3, which were transcriptionally deregulated in HD mouse striatal neurons (Fig. 7a, b and Supplementary Fig. 7c, d). This suggests that epigenetic mechanism might trigger early stress response in HD striatal neurons, which is then reduced upon aging, possibly promoting entry into symptomatic phase. Consistently, M4 genes were downregulated in striatal tissue of symptomatic HD KI mice (e.g., at 6 and 10 months) (Fig. 7c). At 2 months, corresponding to prodromal phase, M4 genes did not appear upregulated in the striatum of HD KI mice, possibly due to bulk analysis of striatal tissue, masking cell-type-specific effects (Fig. 7c). Downregulation of M4 genes was specific to dSPN, at least in 6-month-old HD KI mice, suggesting greater contribution of dSPN to loss of homeostatic response in HD (Fig. 7d). Interestingly, the anti-senescence transcription factor *Glis1*[20] was among M4 genes (Fig. 7b). In WT striatal neurons, *Glis1* was de-repressed over time, primarily due to reduction of H3K27me3, particularly between 2 and 6 months of age (Fig. 7e, f). Age-dependent de-repression of *Glis1* was suppressed in HD KI mice, resulting in decreased expression of *Glis1* when compared to WT upon aging (Fig. 7e, f), suggesting *Glis1*-mediated anti-aging response is attenuated as HD striatal neurons age. Glis1 is lowly expressed in differentiated cells. Interestingly, it is particularly low in SPN vs astrocytes and, within SPN, it is reduced in iSPN vs dSPN (Fig. 7g). Thus, there is a degree of correlation between *Glis1* cellular levels and vulnerability to the HD gene. Further, *Glis1* was specifically downregulated in dSPN of HD mice (Fig. 7g, h), and GLIS1 protein was significantly decreased in HD mouse striatum (Fig. 7i). Together, this raises the intriguing possibility that low physiological levels of *Glis1* (lower in iSPN vs dSPN) and suppression of *Glis1* de-repression during aging (greater in dPSN vs iSPN) contribute to high vulnerability of iSPN and dSPN to the HD gene, respectively, notably preventing efficient anti-aging response. Collectively, our network-based analysis identifies stress response module, including the anti-aging *Glis1* gene, that is abnormally regulated in HD vulnerable neurons.

### Discussion

Here we have performed cell-type-specific and temporal histone modification profiling in HD mice. Our data show accelerated erosion

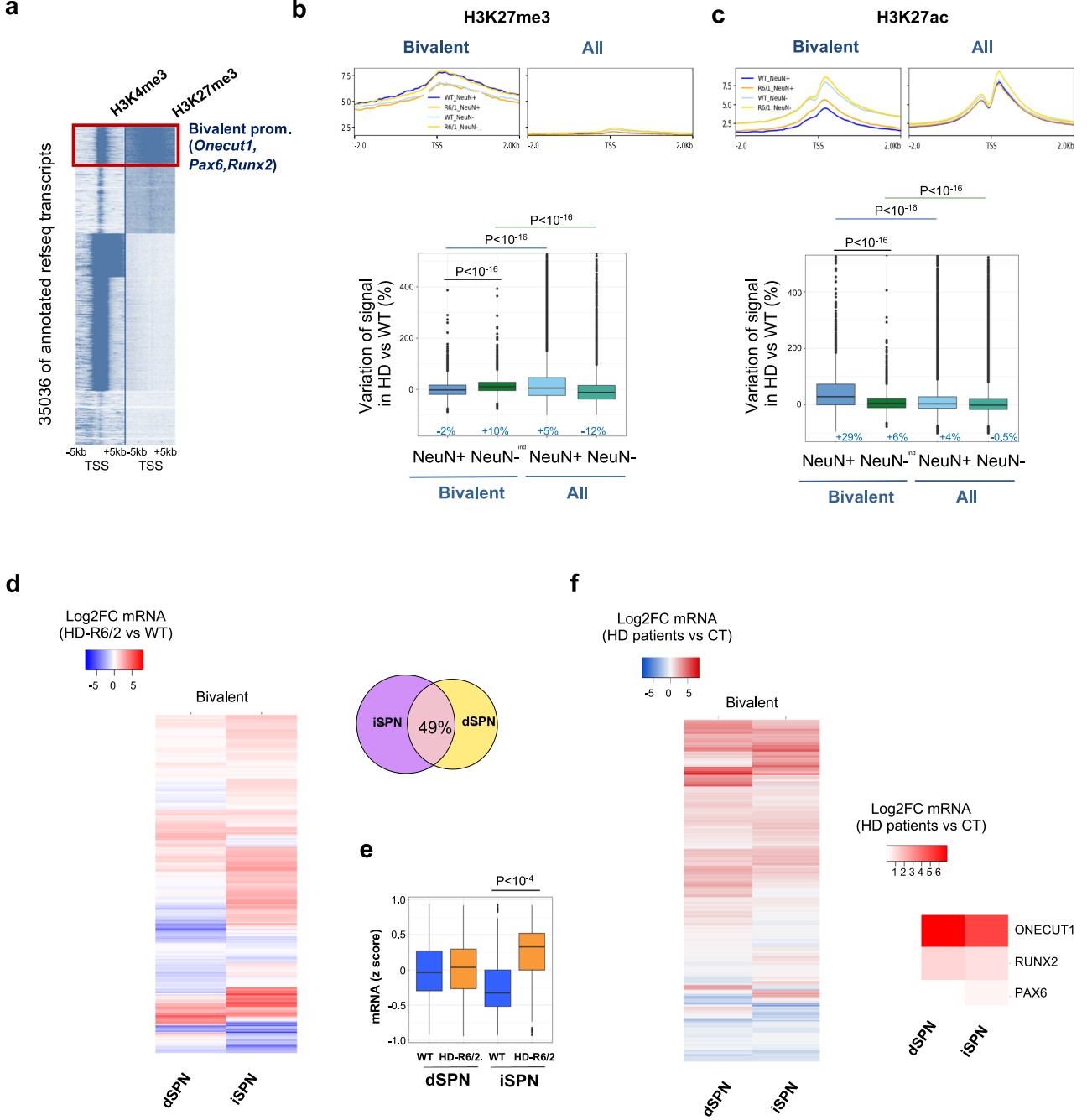

**Fig. 3 | Bivalent promoters are de-repressed in striatal neurons of HD mice.**
**a** Heatmap of the 36,873 annotated mm10 RefSeq gene transcripts, integrating H3K4me3 and H3K27me3 gene profiles from striatal ChIPseq data, showing five distinct epigenetic profiles generated by kmeans clustering. TSS, transcription start site; TTS, transcription termination site. The red box highlights the cluster containing bivalent promoters, including *Onecut1*, *Pax6* and *Runx2*. Top, Metaprofiles showing H3K27me3 (**b**) and H3K27ac (**c**) mean signals in WT (blue) and R6/1 (orange) NeuN+ and NeuN- samples at bivalent and all promoters. Bottom, boxplots showing H3K27ac and H3K27me3 signal variations, expressed as percentage, in R6/1 *vs* WT at bivalent and all promoters. Statistical analysis was performed using Kruskal–Wallis test and Bonferroni correction for multiple testing. Boxplots show median, first quartile (Q1), third quartile (Q3) and range (min, Q1-1.5*(Q3-Q1); max, Q3+1.5*(Q3-Q1). Median values are indicated in blue. H3K27me3, $N = 2$ biological

replicates in each group; H3K27ac, $N = 2$ biological replicates in each group. **d** Left, Heatmap of Log2FC expression values of bivalent genes in R6/2 vs WT dSPN and iSPN (RNAseq data by ref. 11). Right, overlap between upregulated bivalent genes in R6/2 vs WT in iSPN and dSPN. 49% of increased bivalent genes in R6/2 dSPN are increased in R6/2 iSPN. **e** Boxplot showing z-scores of expression values for bivalent genes in dSPN and iSPN in R6/2 and WT. RNAseq data by ref. 11; dSPN, $N = 7$ biological replicates in each group; iPSN, $N = 4$ biological replicates in each group. Boxplots show median, first quartile (Q1), third quartile (Q3) and range (min, Q1-1.5*(Q3-Q1); max, Q3+1.5*(Q3-Q1). Statistical analysis was performed using Kruskal–Wallis test and Bonferroni correction for multiple testing. **f** Left, heatmap of Log2FC expression values of bivalent genes of HD patients vs control individuals in dSPN and iSPN (RNAseq data by ref. 17). Right, zoom showing ONECUT1, RUNX2 and PAX6. Source data are provided as a source data file.

of epigenetic information in HD vulnerable neurons, promoting a loss of cellular identity. Particularly, developmental genes undergo accelerated de-repression in HD striatal neurons, which results from depletion of PcG protein-mediated histone marks (e.g., H3K27me3 and

H2AK119ub) and gain of histone acetylation (e.g., H3K9ac, H3K18ac, H3K27ac). Our data provide additional insight into the mechanism underlying accelerated de-repression of developmental genes in HD striatal neurons, showing paralog switching between PRC1-CBX genes

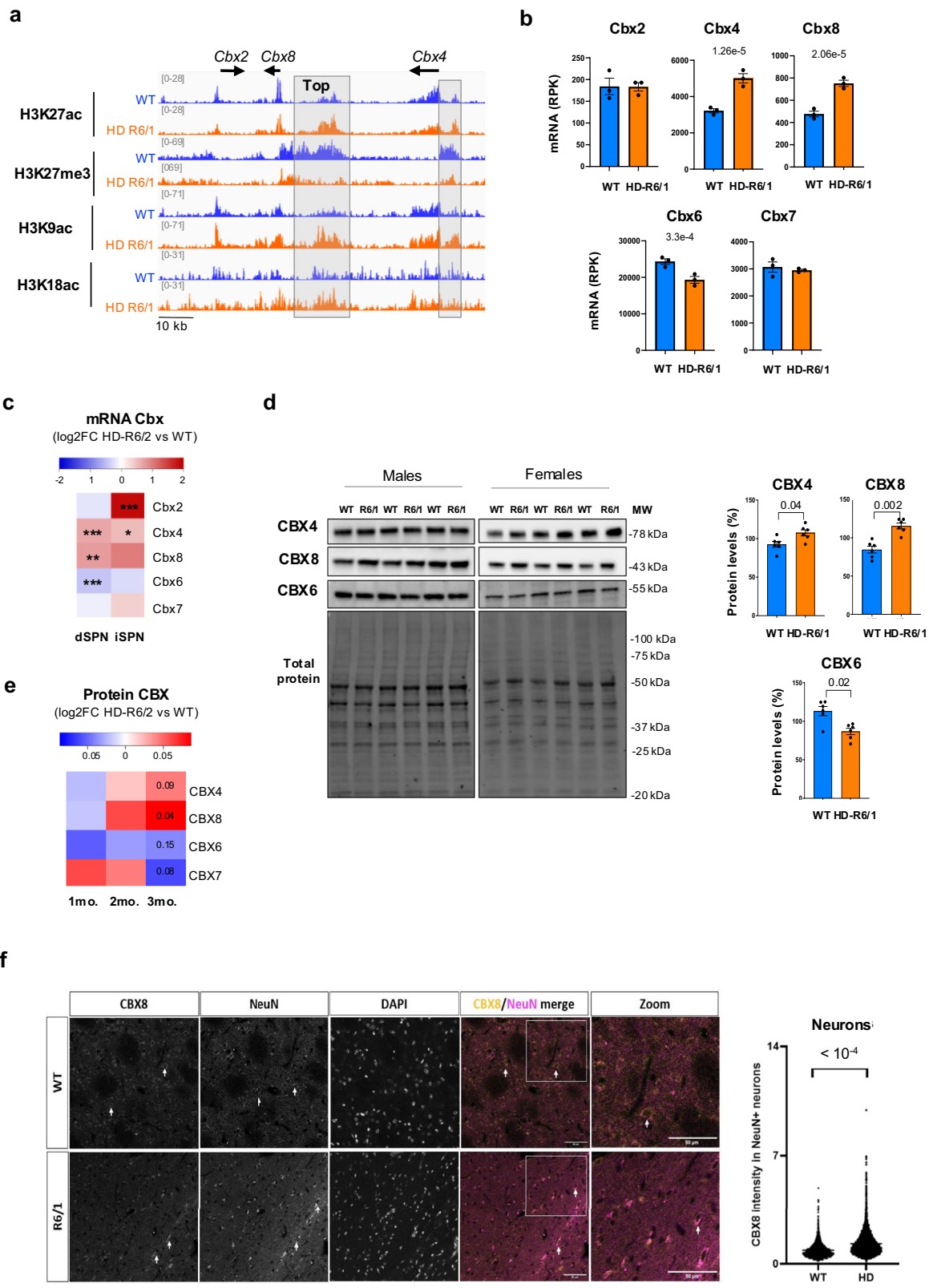

that promotes PRC1-CBX stoichiometry characteristic of immature neurons. Noticeably, this is the first study implicating PRC1 in epigenetic aging. Furthermore, we defined H3K27ac/H3K27me3-based euchromatin score, which applied to subcluster of bivalent developmental transcription factors can be used as an aging index. Euchromatin score further revealed abnormal trajectory of stress response genes over time in HD striatal neurons and identified anti-aging GLIS1

factor that might contribute to both intrinsic and HD gene-mediated vulnerability of SPN. Thus, we propose that altered regulation of stress response in HD vulnerable neurons triggers their accelerated epigenetic aging and dysfunction.

Previous transcriptomic studies showed that downregulated genes in the striatum of HD patients and mouse models are enriched in SPN identity genes, suggesting cellular identity of vulnerable neurons

**Fig. 4 | PRC1-CBX proteins undergo paralog switch in striatal neurons of HD mice. a** IGV genome browser capture showing H3K27ac, H3K27me3, H3K9ac and H3K18ac signals in R6/1 and WT NeuN+ and NeuN- samples at *Cbx2/4/8* genomic locus. The grey boxes highlight significantly H3K27me3-depleted and H3K27ac-enriched regions in R6/1 vs WT NeuN+ samples. The enhancer region between *Cbx4* and *Cbx8* was top significantly H3K27me3-depleted and H3K27ac-enriched region. **b** mRNA levels of Cbx2/4/6/7/8 in bulk striatal tissue of R6/1 and WT mice (RNAseq data and statistics by ref. [74]; *N* = 3 biological replicates in each group). mRNA levels, reads per kilobases (RPK). Mean values +/- sem are shown. Statistics show adj. *P*val, multiple testing correction was performed using the Benjamini–Hochberg method upon analysis of RNAseq data. **c** Heatmap of Log2FC (R6/2 / WT) expression values of PRC1 Cbx paralogs in dSPN and iSPN (RNAseq data by ref. [11]). Statistics show adj. *P*val, multiple testing correction was performed using the Benjamini–Hochberg method upon analysis of RNAseq data. *, adj. *P*val < 0.05; **, adj. *P*val < 0.01; ***, adj.

*P*val < 0.001. **d** Left, Immunoblots showing CBX4, CBX6, CBX8 and total proteins levels in the striatum of R6/1 and WT mice. Male and female samples are specified. MW, molecular weight. Right, bargraphs showing CBX4, CBX6 and CBX8 protein levels. Values were normalized to total protein. Mean values +/- sem are shown. Statistical analysis was performed using Mann–Whitney test (two-sided). *N* = 6 biological replicates in each group. **e** Heatmap of Log2FC PRC1-CBX protein level values in striatal tissue of HD-R6/1 vs WT mice of 1, 2 and 3 months (proteomic data, https://www.ebi.ac.uk/pride/archive/projects/PXD013771 and https://www.hdinhd.org/). **f** Representative images of CBX8 co-stained with neurons (NeuN) and DNA (DAPI) in the striatum region of 15 week-old WT and R6/1 mice (left panel), arrows indicate double-positive cells (CBX8/NeuN), with quantification of CBX8 intensity in neuron (right panel, *N* = 5 animals per group; WT, 4920 neurons count; R6/1, 5906 neurons count). Source data are provided as a source data file.

is compromised in HD[10–12,21–23]. Epigenomic studies supported epigenetically-driven mechanism since H3K27ac was depleted at SPN identity genes[8,9]. Here, our cell-type-specific and temporal profiling showing progressive euchromatinization and transcriptional upregulation of developmental genes in HD mouse striatal neurons support the view that loss of identity of HD SPN results from both repression of identity genes and de-repression of developmental genes. Due to the scarcity of human data, particularly the lack of temporal epigenomic and transcriptomic data, it is difficult to extend the results to humans. Nonetheless, recent transcriptomic data suggest that the mechanism occurs in humans. First, using published FANS-seq data[24], we found that bivalent genes, including *ONECUT1*, *PAX6* and *RUNX2*, were upregulated in SPN of HD patients. Second, simultaneous single-cell analysis of CAG expansion and transcriptome shows that long CAG expansions that result from somatic instability induce deleterious transcriptional erosion in SPN of HD patients[25]. This erosion is similar to that observed in HD mice: it is characterized by progressive downregulation of SPN identity genes as well as de-repression of developmental genes normally repressed, so-called de-repression crisis. Cellular aging leads to erosion of epigenetic information, which results in repression of cellular identity genes and de-repression of developmental genes and drive a loss of cellular identity and function[7,26]. This raises the exciting hypothesis that neuronal vulnerability of SPN in HD is tightly linked to propensity for epigenetic erosion. The conclusion might extend to cortical neurons since transcriptomic study shows that developmental genes, including *ONECUT1* and *PAX6*, are upregulated in vulnerable neurons of layer 5a but not in the resilient Betz cells in post-mortem brains of HD patients[27]. Strikingly, similar mechanism might occur in Alzheimer disease (AD). Transcriptomic and chromatin accessibility changes associated with AD showed epigenetic erosion and loss of cellular identity in AD-affected brain cells[28]. Particularly, chromatin accessibility and expression of developmental genes was increased in AD cortical neurons[28]. Though time course analysis is lacking with those human data, this suggests AD-vulnerable neurons undergo accelerated epigenetic aging, and that perhaps common epigenetic underpinning contribute to neuronal vulnerability in HD and AD.

The quantification of cellular aging using epigenetic clocks essentially relies on bulk tissue analysis of few DNA methylation sites[29]. Such a DNA clock was used to show that epigenetic aging is accelerated in the cortex of HD patients[30]. However, DNA clocks are universal clocks, which do not capture cell-type-specific changes. Moreover, they lack functional readout, which limits the understanding of the biology of aging. Recently, cell-type-specific transcriptomic clocks were developed to measure the contribution of different brain cells to brain aging[31]. Here we show that cell-type-specific histone modifications (i.e., H3K27ac and H3K27me3) can also be used as a proxy to measure epigenetic age. Specifically, we show that euchromatin score at developmental transcription factors, which takes into consideration H3K27ac and H3K27me3 signals, can precisely measure acceleration of

epigenetic aging in HD striatal neurons. Future studies might determine whether the approach could be used to build histone modification-based epigenetic clock.

It is believed that bivalent chromatin state, defined as the simultaneous presence of H3K4me3 and H3K27me3, poises important regulatory genes for expression or repression during cell-lineage specification[32,33]. Gene bivalency is also observed in differentiated cells, including striatal neurons, though functional relevance of bivalent chromatin state in mature cells remains elusive[33,34]. Consistent with our results, PRC2 deficiency in mouse striatal neurons led to de-repression of bivalent developmental transcription factors as well as progressive neurodegeneration, which is reminiscent to HD[34]. This supports the view that preserving bivalent chromatin state at developmental genes is critical to maintaining cellular identity and preventing cellular aging.

HTT interacts with PRC2 and was found to control bivalent gene regulation during neural differentiation[35]. Thus, it is possible that mutant HTT enhances accelerated de-repression of developmental genes via direct mechanism. However, de-repressed developmental transcription factors in HD striatal neurons also showed PRC1-dependency. This is consistent with results showing that up to 40% of bivalent genes are marked with both H3K27me3 and H2AK119ub in embryonic stem cells and other cells and that co-marked genes are enriched in genes encoding developmental transcription factors[33,36]. Supporting an implication of PRC1 in accelerated loss of cellular identity of HD striatal neurons, we show progressive *Cbx6/7* to *Cbx2/4/8* paralog switching, promoting PRC1-CBX stoichiometry reminiscent to that associated with neural differentiation[16,18]. PRC1-CBX proteins bind H3K27me3 and induce chromatin compaction, inducing phase separation-driven molecular condensates, thanks to intrinsically disordered region (IDR)[16,37–40]. However, PRC1-CBX stoichiometry determines condensate formation, due to distinct phase separation capacities of the different PRC1-CBX proteins. In cells, CBX2/4/7/8 appear as defined condensates, though CBX7, lacking IDR, shows low chromatin compaction activity[41–43]. In contrast, CBX6 does not form condensates in cells[16,40]. Further, in vitro study supports a model where PRC1-CBX2 acts as a scaffold initiating PRC1 condensate formation, which can then recruit PRC1-CBX4/7/8 but not PRC1-CBX6[40]. CBX6 is the predominant PRC1-CBX isoform in SPN. Thus, an attractive hypothesis is that changes in PRC1-CBX stoichiometry in HD vulnerable neurons affects phase separation and chromatin compaction activities, which triggers epigenetic reprogramming that promotes loss of cellular identity. Additionally, PRC1-CBX changes might play more direct role in de-repression of developmental genes in HD striatal neurons. Strikingly, CBX4 can act as a transcriptional activator, which recruits the histone acetyltransferase GCN5 and promotes histone acetylation, including H3K27ac, notably at the promoter of *Runx2*, a transcription factor implicated in skeletal development[44–46]. Interestingly, our data show that *Runx2* undergoes accelerated epigenetic de-repression in HD striatal neurons, particularly at early stage of the

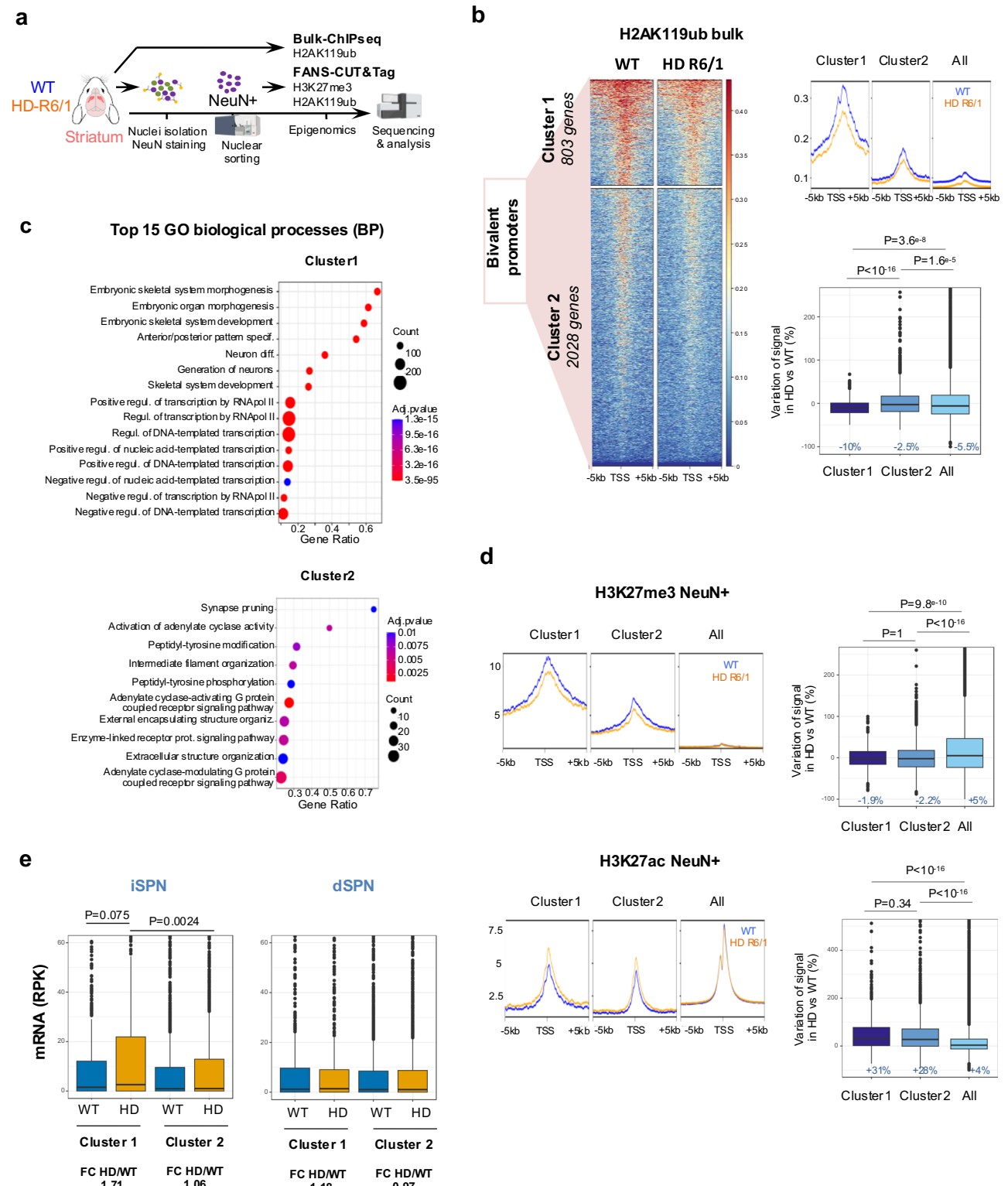

pathology. Since RUNX2 is a pioneer transcription factor, this might initiate de-repression of self-regulating developmental transcription factors, which drives positive autoregulatory feedback loop leading to de-repression crisis. In fact, such feedback loop driving re-activation of bivalent genes, including Cbx4 and autoregulatory transcription factors such as *Onecut1*, was described in striatal neurons deficient in PRC2[34]. Intriguingly, this associated with repression of striatal neuron identity genes[34]. These results further highlight critical role for polycomb repressive proteins in epigenetic erosion of striatal neuron

identity, though their precise interplay needs to be addressed in the future.

The information theory of aging posits that DNA repair mechanism is the driving force[6,7]. However, the mechanism that links DNA repair and epigenetic aging is unknown. PRC1, which coordinates DNA repair and transcriptional machineries, might be a missing link[47,48]. Specifically, PRC1 induces chromatin compaction and transcriptional silencing at damaged genes and promotes DNA repair[48,49]. Interestingly, human and mice studies have linked DNA repair and HD[17,50–54].

**Fig. 5 | H2AK119ub is depleted at subcluster of bivalent promoters in the striatum of HD mice. a** Scheme illustrating bulk ChIPseq and FANS-CUT&Tag experiments conducted on 15–20 week-old R6/1 and WT mice (Created in BioRender. Boutillier, A. (2025) https://BioRender.com/x74t675). H2AK119ub bulk ChIPseq, $N = 2$ biological replicates; H2AK119ub and H3K27me3 FANS-CUT&Tag, $N = 3$ biological replicates. **b** Left, kmeans clustering analysis of bivalent promoters using striatal H2AK119ub ChIPseq data identified H2AK119ub-high subcluster (cluster 1) and H2AK119ub-low subcluster (cluster 2). Top right, metaprofile showing H2AK119ub signals in R6/1 and WT samples at cluster 1, cluster 2 and all promoters. Bottom right, boxplots showing H2AK119ub signal variations, expressed as percentages, in R6/1 *vs* WT at cluster 1, cluster 2 and all promoters. Boxplots show median, first quartile (Q1), third quartile (Q3) and range (min, Q1-1.5*(Q3-Q1); max, Q3+1.5*(Q3-Q1). Median values are indicated in blue. Statistical analysis was performed using Kruskal–Wallis test and Bonferroni correction for multiple testing. **c** Top 15 gene ontology (GO) processes (biological processes, BP) enriched in

cluster 1 and cluster 2. **d** Metaprofiles showing H3K27me3 (Top left) and H3K27ac (Bottom left) mean signals in WT (blue) and R6/1 (orange) NeuN+ ChIPseq samples at cluster 1, cluster 2 and all promoters. Boxplots showing H3K27me3 (Top right) and H3K27ac (Bottom right) signal variations in R6/1 *vs* WT at cluster 1, cluster 2 and all promoters. Boxplots show median, first quartile (Q1), third quartile (Q3) and range (min, Q1-1.5*(Q3-Q1); max, Q3+1.5*(Q3-Q1). Median values are indicated in blue. Statistical analysis was performed using Kruskal–Wallis test and Bonferroni correction for multiple testing. **e** Boxplots showing mRNA levels of genes in cluster 1 and cluster 2 in iSPN and dSPN of R6/2 and WT mice (RNAseq data by ref. 11). Fold change (FC) of HD/WT values are shown. mRNA levels, reads per kilobases (RPK). Boxplots show median, first quartile (Q1), third quartile (Q3) and range (min, Q1-1.5*(Q3-Q1); max, Q3+1.5*(Q3-Q1). Statistical analysis was performed using Kruskal–Wallis test and Bonferroni correction for multiple testing. Source data are provided as a source data file.

First, GWAS studies show that HD modifiers are enriched in DNA repair genes, notably genes that modulate somatic CAG instability[52]. Second, Huntingtin (HTT), the protein mutated in HD, is implicated in DNA damage response (DDR) and mutated HTT impairs DDR[55,56]. Third, mutated HTT progressively forms toxic nuclear aggregates, which can recruit chromatin regulators and modify their activities, though it is yet unclear whether this includes PRC1[57–60]. Nonetheless, PRC1 is a good candidate to convey HD-associated DNA-damage/repair mechanisms to erosion of epigenetic information in vulnerable neurons.

Finally, we identified module enriched in stress response genes showing increased euchromatin score at prodromal disease stage and then epigenetically repressed at symptomatic stage in striatal neurons of HD mice. This is in line with studies showing that neurodegenerative conditions lead to impaired stress response and homeostasis regulation[7]. Further, shape deformation analysis support loss of transcriptional regulation of homeostatic responses over time in HD striatal neurons[61]. Our data suggest epigenetic underpinning. Noticeably, the stress response module comprised the anti-aging gene *Glis1*. GLIS1 enhances somatic cell reprogramming when expressed with the Yamanaka factors OCT4, SOX2 and KLF4 (OSK)[62]. Interestingly, GLIS1 was sufficient to reprogram senescent cells into pluripotent cells, and transient expression of OSK factors could rejuvenate aged cells, reverting epigenetic erosion and loss of cellular identity[7,20,63]. Thus, GLIS1 might counteract epigenetic erosion in aging cells. HD transcriptomic data support a role for GLIS1 in SPN vulnerability: *Glis1* was moderately expressed in dSPN and reduced in HD vs WT conditions, while it was low expressed in both HD and WT iSPN, suggesting that GLIS1 contributes to both disease-associated and intrinsic vulnerability mechanisms. Collectively, our data provide new conceptual frame to fight HD, highlighting major role of epigenetic aging in the disease.

## Methods
### Animals
Heterozygous Q140 KI mice and heterozygous R6/1 mice were maintained on C57BL/6J genetic background. All animal studies were conducted in accordance with French regulations (EU Directive 2010/63/UE–French Act Rural Code R 214-87 to 126). The animal facility was approved by veterinary inspectors (authorization no. E6748213) and complies with the Standards for Human Care and Use of Laboratory Animals of the Office of Laboratory Animal Welfare. All procedures were approved by local ethics committee (CREMEAS) and French Research Ministry (no. APAFIS#31527_2021051713407251v3, APAFIS#11532-2017092618102093v7 and no. APAFIS#504-2015042011568820_v3). Mice were housed in a controlled-temperature (22 °C) and -humidity (40%) room maintained on a 12 h light/dark cycle. Food and water were available ad libitum. Genotyping was performed by PCR, using ear-skin DNA obtained from 1-month-old mice with primers amplifying the CAG repeat region

within the exon 1 of the Huntingtin gene. For molecular analyses, mice were killed by cervical dislocation and their striata were rapidly dissected, snap-frozen and stored at −80 °C. Both male and female animals were used in the experiments, and WT littermates were used as controls.

### Fluorescence activated nuclear sorting - crosslinked nuclei (ChIP)
Cell-type specific nuclear purification was performed using fluorescent-activated nuclear sorting, as described[9]. Briefly, frozen striatal tissue from WT and R6/1 animals was homogenized in ice-cold PBS supplemented with 1× Protease Inhibitors Cocktail (PIC, cOmplete EDTA free, Roche) and cross-linked in 1% formaldehyde for 15 min at room temperature. Cross-linking was stopped by the addition of glycine to final concentration 0.125 M and tissue was washed using ice-cold PBS. Cells were then lysed in Cell Lysis Buffer (10 mM Hepes pH8; 85 mM KCl; 0.5% NP-40) and nuclei were collected after treatment with Nuclear Extraction Buffer (0.5% SDS, 10 mM EDTA pH8, 50 mM Tris). Purified nuclei were then resuspended in PBTB (PBS 1×, 5% BSA, 0.5% Tween-20) + 1× PIC, 3% Normal Horse Serum (NHS) and stained using NeuN antibody (1:1000, Merck Millipore, MAB377). After washing, nuclei were labeled with Alexa Fluor 488 donkey anti-mouse IgG antibody (1:1500) and washed with ice-cold PBS. Immunostained nuclei were sorted using BD Aria Fusion flow cytometer, recovered in ice-cold 1× PBS, pelleted and stored at −80 °C for posterior ChIP-seq experiments.

### Fluorescence activated nuclear sorting - non crosslinked nuclei (CUT&Tag)
Briefly, frozen striata were pulverized using a grinder and pestle settle on dry ice and reconstituted in PBS 1× supplemented with 1× PIC. Cell lysis and nuclear extraction were performed by a 10-min incubation in LB1 buffer (50 mM HEPES pH7.5, 140 mM NaCl, 1 mM EDTA pH8, 10% glycerol, 0.5% NP-40, 0.25% Triton X-100, 1× PIC) and mechanical dissociation in a glass douncer. Nuclei were then filtered with a 50-μm pore size cell strainer (Sysmex Celltrics) and stained using NeuN antibody conjugated to AlexaFluor 405 (1:200, Novus Biologicals, 1B7 AF405) or AlexaFluor 647 (1:500, Abcam, EPR12763). Nuclear suspension was then sorted using BD FACS ARIA II flow cytometer.

### Chromatin Immunoprecipitation (ChIP)
FANS-ChIPseq. NeuN+ and NeuN- sorted nuclei from 15 to 20-week-old R6/1 mice and WT littermates were used. Chromatin was extracted from nuclei by sonication and divided to allow immunoprecipitation of the same extract with H3K27ac, H3K27me3 antibodies and Input controls (1 million nuclei per ChIP were used). ChIPseq data were replicated through two independent experiments. ChIPseq was performed as described[9] using antibodies targeting H3K27ac (ab4729, Abcam) and H3K27me3 (C15410195, diagenode). Briefly, nuclei were suspended in Sonication buffer (50 mM HEPES pH7.9, 140 mM NaCl,

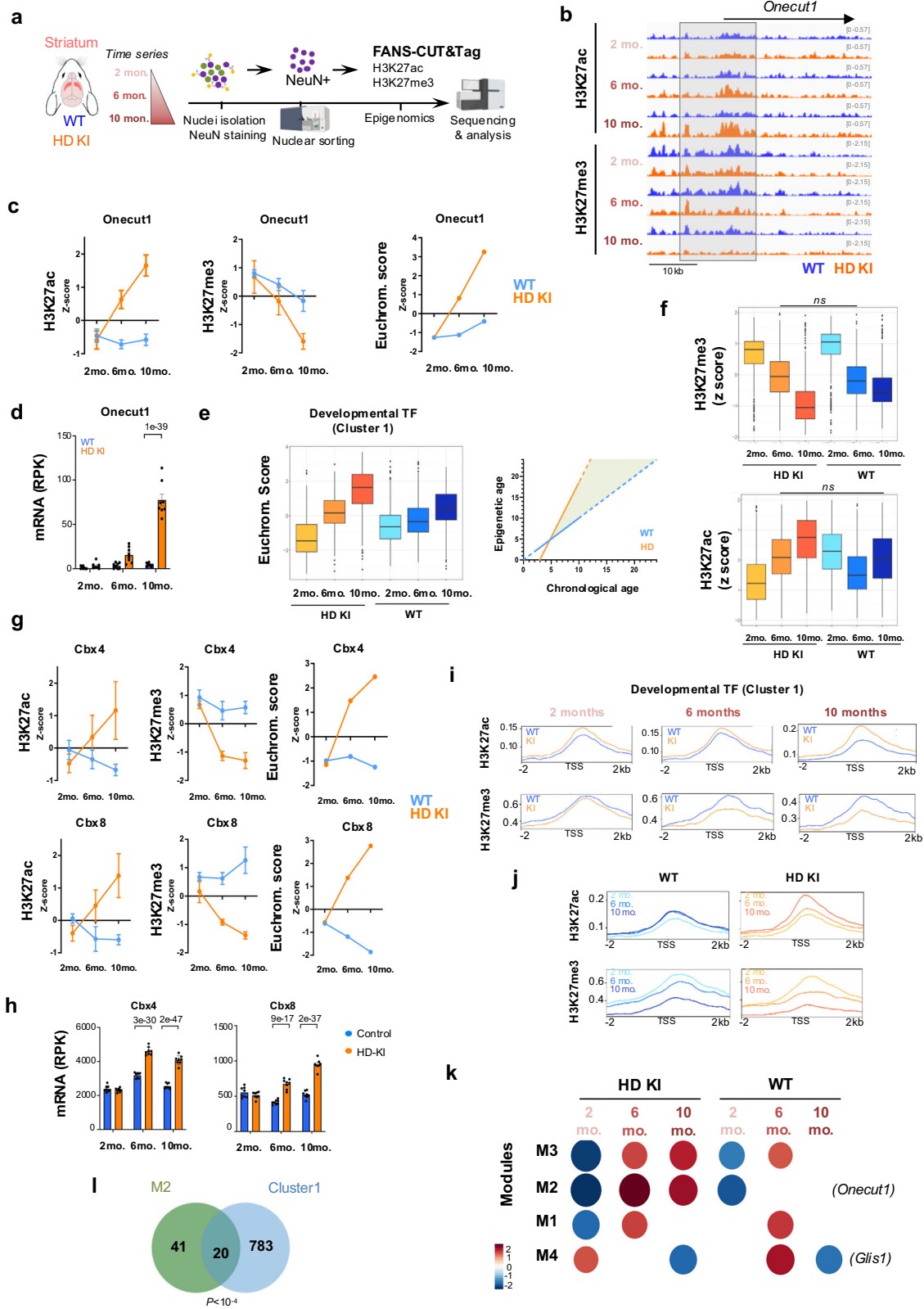

1 mM EDTA, 1% Triton X-100, 0.1% NaDOC, 0.1% SDS, 1x PIC) and sonicated to obtain DNA fragments <500 bp using Covaris E220 Ultrasonicator (Covaris). The soluble chromatin fraction was pretreated with protein A Agarose/Salmon Sperm DNA (16-157, Sigma-Aldrich) for 45 min at 4 °C. Subsequently, samples were incubated overnight at 4 °C with corresponding primary antibodies. Protein A Agarose/Salmon Sperm DNA was then added and the mixture was

incubated for 3 h at 4 °C in a shaker. Agarose beads were washed, protein–DNA complexes were eluted from the beads and de-crosslinked overnight with RNAse A at 65 °C. Proteins were eliminated by 2 h incubation at 45 °C with Proteinase K, and DNA recovered using Qiagen MiniElute PCR Purification Kit.

Bulk ChIPseq. Striatal tissues from 15 to 20-week-old R6/1 mice and WT littermates were used. Two biological replicates were

**Fig. 6 | De-repression of developmental genes is an aging signature accelerated in striatal neurons of HD mice. a** Scheme illustrating temporal FANS-CUT&Tag experiments conducted on HD Q140 knockin (HD KI) mice and WT mice of 2, 6 and 10 months of age (Created in BioRender. Boutillier, A. (2025) https://BioRender.com/t46w032). H3K27ac and H3K27me3 FANS-CUT&Tag at 2 months, $N = 3$ biological replicates; H3K27ac and H3K27me3 FANS-CUT&Tag at 6 months, $N = 3$ biological replicates; H3K27ac and H3K27me3 FANS-CUT&Tag at 10 months, $N = 2$ biological replicates. **b** IGV genome browser capture showing H3K27ac and H3K27me3 signals at *Onecut1* locus in HD KI and WT NeuN+ samples produced using 2, 6 and 10 month-old mice. **c** Plots showing z-score values for H3K27ac, H3K27me3 and euchromatin score for *Onecut1* gene, in 2, 6 and 10 month-old HD KI and WT NeuN+ samples. Mean values +/- sem are shown. **d** mRNA levels of *Onecut1* in bulk striatal tissue of 2, 6 and 10 month-old HD Q140 KI and WT mice (RNAseq data by ref. 10; $N = 10$ biological replicates in each group). mRNA levels, reads per kilobases (RPK). Mean values +/- sem are shown. Statistics show adj. *P*val, multiple testing correction was performed using the Benjamini–Hochberg method upon analysis of RNAseq data. **e** Left, Boxplot showing euchromatin score for cluster 1 genes (i.e., developmental transcription factors -TF-) in HD KI and WT NeuN+ samples that were generated using 2, 6 and 10 month-old mice (left). Boxplots show median, first quartile (Q1), third quartile (Q3) and range (min, Q1-1.5*(Q3-Q1); max, Q3+1.5*(Q3-Q1). Statistical analysis was performed using Kruskal–Wallis test and Bonferroni correction for multiple testing; all pairwise comparisons are statistically different ($p < 0.05$). Right, linear regression computed using cluster 1 mean

euchromatin score shows distortion between chronological and epigenetic ages in striatal neurons of HD KI mice (green area), which reflects accelerated epigenetic aging. **f** Boxplots showing H3K27ac (top) and H3K27me3 (bottom) z-scores for cluster 1 genes in HD KI and WT NeuN+ samples generated using 2, 6 and 10 month-old mice. Boxplots show median, first quartile (Q1), third quartile (Q3) and range (min, Q1-1.5*(Q3-Q1); max, Q3+1.5*(Q3-Q1). Statistical analysis was performed using Kruskal–Wallis test and Bonferroni correction for multiple testing; pairwise comparisons are statistically different ($p < 0.05$), except when indicated (ns). **g** Plots showing z-score values of H3K27ac, H3K27me3 and euchromatin score for *Cbx4* and *Cbx8* genes, in 2, 6 and 10 month-old HD KI and WT NeuN+ samples. Mean values +/- sem are shown. **h** Mean mRNA levels of *Cbx4* and *Cbx8* in bulk striatal tissue of 2, 6 and 10 month-old HD Q140 KI and WT mice (RNAseq data by ref. 10; $N = 10$ biological replicates in each group). mRNA levels, reads per kilobases (RPK). Mean values +/- sem are shown. Statistics show adj. *P*val, multiple testing correction was performed using the Benjamini–Hochberg method upon analysis of RNAseq data. **i** Metaprofiles showing H3K27ac and H3K27me3 mean signals at cluster 1 gene promoters in HD KI vs WT NeuN+ samples prepared using 2, 6 and 10 month-old mice. **j** Above H3K27ac and H3K27me3 metaprofiles were grouped according to age. **k** Gene set enrichment analysis showing showing module normalized enrichment score (NES) identified by CEMiTool analysis using euchromatin score. M2 and M4 modules include *Onecut1* and *Glis1*, respectively. **l** Overlap between M2 (green) and cluster1 (blue) genes. Statistics of overlap (*P*) was assessed using a binomial test. Source data are provided as a source data file.

generated. ChIPseq was performed as described[9] using antibody targeting H2AK119ub (8240, cell signaling). Briefly, pooled tissues were cut into small fragments, fixed in 1% formaldehyde and incubated for 15 min at room temperature. Cross-linking was stopped by the addition of glycine to final concentration 0.125 M. Tissue fragments were washed with cold PBS supplemented with protease inhibitors. The tissues were then mechanically homogenized in sonication buffer to obtain a homogeneous solution, then the protocol was the same as the one described for sorted nuclei, starting by sonication using Covaris E220 Ultrasonicator (Covaris).

### FANS-CUT&Tag
NeuN+ sorted nuclei from 15- to 20 week-old R6/1 mice and WT littermates were used. Sorted nuclei preparation was divided to allow immunoprecipitation of the same extract with H3K9ac (1:50, Ab4441, Abcam, $N = 4$–5 biological replicates) and H3K18ac (1:50, C15410139, Diagenode, $N = 2$ biological replicates), and immunoprecipitation of H3K27me3 (C15410195, diagenode, $N = 3$ biological replicates) and H2AK119ub (1:50, 8240, cell signaling, $N = 3$ biological replicates) and IgG control (1:50, C15410206, Diagenode) with 50,000 nuclei/CUT&Tag. For temporal CUT&Tag experiment on Q140 KI mice and WT littermates, NeuN+ sorted nuclei from 2, 6, 10-month-old mice were used. Three biological replicates were generated. Sorted nuclei preparations were divided to allow immunoprecipitation of the same extract with H3K27ac (1:300, ab4729, Abcam), H3K27me3 (1:50, C15410195, diagenode) and IgG control (1:50, C15410206, Diagenode), using 50,000 nuclei/CUT&Tag. CUT&Tag was essentially performed as described[14]. Briefly, concanavalinA-coated beads (93569S, Cell signaling) were added to nuclei suspension at room temperature under rotation to attach nuclei. Then corresponding primary antibodies were incubated overnight at 4 °C. Beads-nuclei complex were washed and incubated 1 h at RT with proteinA-Transposase5 complex (C01070001, Diagenode) to a final concentration of 1:250. Tagmentation was done by incubating samples 1 h in MgCl2 at 37 °C, then proteins were eliminated by 1 h incubation in SDS and Proteinase K at 55 °C. DNA was recovered using Qiagen MinElute PCR Purification Kit.

### Library preparation & sequencing
ChIPseq. ChIP samples were purified using Agencourt AMPure XP beads (Beckman Coulter) and quantified with Qubit (Invitrogen). ChIP-seq libraries were prepared from 2 ng of double-stranded purified DNA

using the MicroPlex Library Preparation kit v2 (C05010014, Diagenode), according to manufacturer's instructions. Illumina-compatible indexes were added through PCR amplification (7 cycles). Amplified libraries were purified and size-selected using Agencourt® AMPure® XP beads (Beckman Coulter) to remove unincorporated primers and other reagents. Prior to analyses, DNA libraries were checked for quality and quantified using a 2100 Bioanalyzer (Agilent). Libraries were sequenced on Illumina Hiseq 4000 sequencer as paired-end or single-end 50 base reads following Illumina's instructions. Image analysis and base calling were performed using RTA 2.7.3 and bcl2fastq 2.17.1.14.

CUT&Tag. CUT&Tag libraries were prepared from the eluted double-strand DNA using NEBNext HiFi 2x PCR master mix (BioLabs) and Illumina compatible indexes (20091654, Illumina Nextera) (13 cycles). Amplified libraries were purified and size-selected using SPRI select beads (B23317, Beckmann Coulter) at 1.3 time, two times consecutively. Prior to sequencing, DNA libraries were checked for quality and quantified using a 2100 Bioanalyzer by Agilent high sensitivity DNA kit. Libraries were sequenced on an Illumina NextSeq 2000 sequencer as paired-end 50 base reads. Image analysis and base calling were performed using RTA version 2.7.7 and BCL Convert version 3.8.4.

### Sequence alignment, peak detection and annotation, differential analysis
ChIPseq. Reads were mapped onto mouse reference assembly GRCm38/mm10 using Bowtie2 v2.4.3. aligner[64]. Reads which mapping quality is below 10 were removed using samtools v1.15.1[65]. Prior to peak calling, biological replicates were pooled using samtools merge v1.15.1. Then, peak calling was done using SICER v1.1[66] with the following parameters: window size: 200; FDR controlling significance: 1e-2, Gap size parameters were determined according to the score value estimated by SICER: selected values of gap sizes are 1000 for H3K27ac and 1400 for H2AK119ub and H3K27me3. Peaks were annotated relative to genomic features using Homer v4.11. Differential enrichment analysis of H2AK119ub ChIPseq data was done using SICER v1.1 with the following parameters: Species: mm10; Effective genome size as a fraction of reference genome: 0.74; Threshold for redundancy allowed for treated reads: 1; Threshold for redundancy allowed for WT reads: 1; Window size: 200 bps; Fragment size: 200 bps. The shift for reads is half of 200. Detected peaks were combined to get the union of all

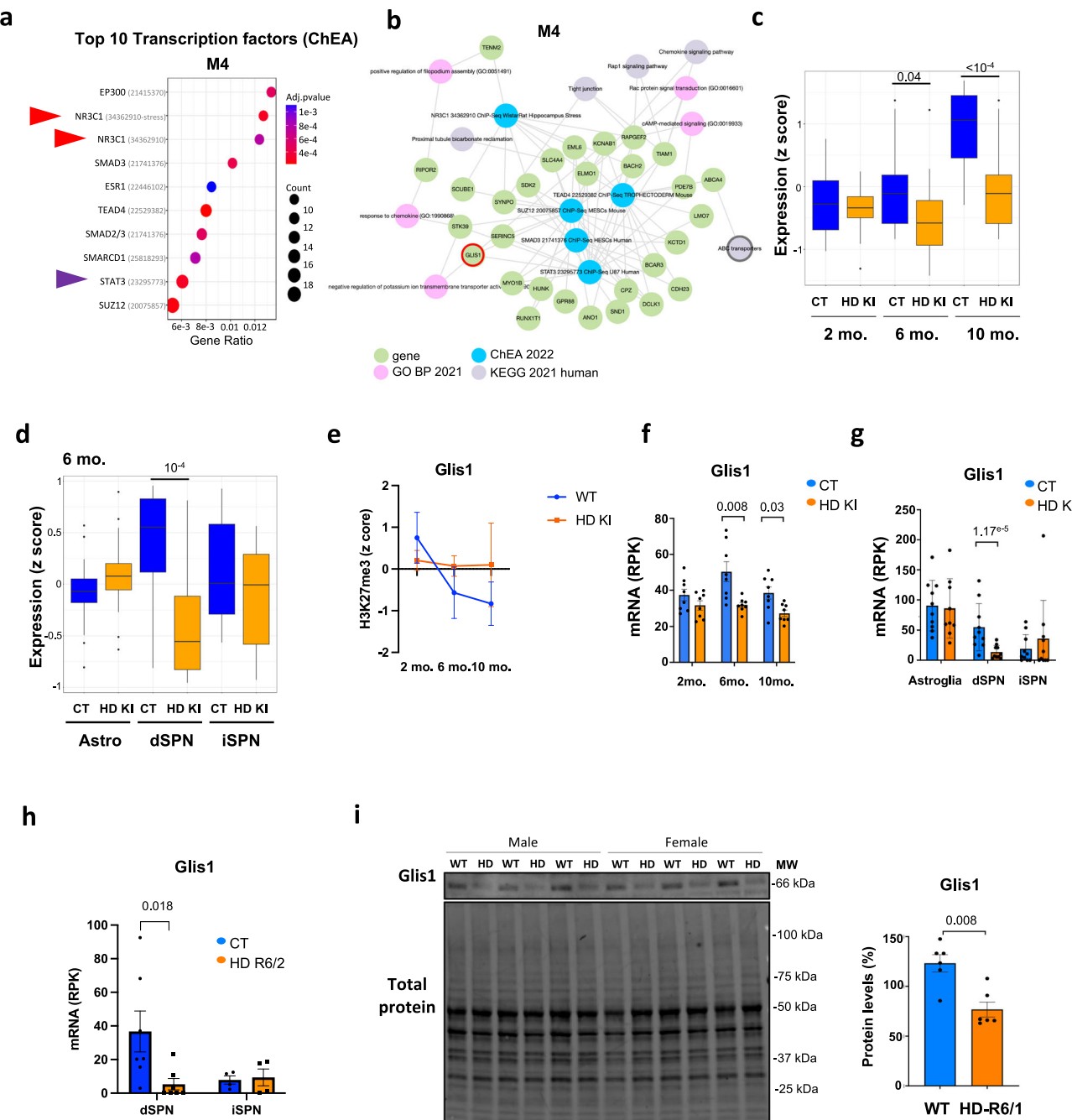

**Fig. 7 | Epigenetic regulation of stress response during aging is abnormal in HD mouse striatal neurons. a** Top 10 predicted transcriptional regulators (ChEA) enriched in M4 module. ChEA terms are shown as a function of gene ratio, gene count and adj. *Pval*. Adj. *Pval* were calculated using the Benjamini−Hochberg method for multiple testing correction **b** Network representation of functional enrichment analysis of M4 genes. Green dots represent M4 genes, blue dots, enriched regulatory transcription factors identified by ChEA analysis, pink dots, enriched BP and grey dots, enriched KEGG pathways. The red-green arrow highlights *Glis1*. **c** Boxplot showing z-scores of expression values of M4 genes in the striatum of HD Q140 KI and WT mice of 2, 6 and 10 months of age (RNAseq data by ref. 10; *N* = 8 biological replicates in each group). Boxplots show median, first quartile (Q1), third quartile (Q3) and range (min, Q1-1.5*(Q3-Q1); max, Q3+1.5*(Q3-Q1). Statistical analysis was performed using Kruskal−Wallis test and Bonferroni correction for multiple testing. **d.** Boxplot showing z-scores of expression values of M4 genes in astroglia, dSPN and iSPN of 6 month-old HD Q175 KI (HD KI) and control (CT) mice. RNA seq data by ref. 11; astroglia, *N* = 10 biological replicates in each group; dSPN, *N* = 10 biological replicates in each group; iPSN, *N* = 10 biological replicates in each group. Boxplots show median, first quartile (Q1), third quartile

(Q3) and range (min, Q1-1.5*(Q3-Q1); max, Q3+1.5*(Q3-Q1). Statistical analysis was performed using Kruskal−Wallis test and Bonferroni correction for multiple testing. **e** Plots showing H3K27me3 z-score values at *Glis1* gene, in 2, 6 and 10 month-old HD KI and WT NeuN+ samples. Mean values +/- sem are shown. **f** mRNA levels of *Glis1* in bulk striatal tissue of 2, 6 and 10 month-old HD Q140 KI (HD KI) and control (CT) mice (RNAseq data by ref. 10). Mean values +/- sem are shown. **g** mRNA levels of *Glis1* in striatal astroglia, dSPN and iSPN of Q175 knockin (HD KI) and control (CT) mice (right, RNAseq data by ref. 11). Mean values +/- sem are shown. **h** mRNA levels of *Glis1* in dSPN and iSPN of HD R6/2 (HD R6/2) and control (CT) mice (right, RNAseq data by ref. 11). mRNA levels, reads per kilobases (RPK). Mean values +/- sem are shown. Statistics show adj. *P*val, multiple testing correction was performed using the Benjamini−Hochberg method upon analysis of RNAseq data. **i.** Left, immunoblots showing Glis1 proteins in the striatum of R6/1 and WT mice. Male and female animals are indicated. MW, molecular weight. Right, bargraph showing quantifications of Glis1 protein levels. Values were normalized to total proteins. Mean values +/- sem are shown. Statistical analysis was performed using Mann−Whitney test (two-sided). Source data are provided as a source data file.

peaks using the tool Bedtools merge v2.30.0[67]. Differential enrichment analysis (H3K27ac and H3K27me3 ChIPseq) was done with DESeq2 bioconductor library (DESeq2 v1.34.0) using R_v3.3.2, and data were normalized using the method proposed by Anders and Huber[68] using read counts per peaks. Then, comparisons of interest were performed using the method proposed by ref. 69. Resulting *p*-values were adjusted for multiple testing using the Benjamini and Hochberg method. Enriched and depleted peaks were defined using adj *p*-value < 10e-5 (SICER) and <0.1 (DESeq2).

CUT&Tag. Data were preprocessed with Cutadapt v4.0 to trim adapter sequences (Nextera Transposase Sequence) from 3′ end of reads. Cutadapt was used with the following parameters '-a CTGTCTCTTATA -A CTGTCTCTTATA -m 25:25'. Reads were mapped to Mus musculus genome (assembly mm10) using Bowtie2 v2.5.0 with default parameters except for "–end-to-end –very-sensitive –no-mixed –no-discordant -I 10 -X 700". Prior to peak calling, biological replicates were pooled using samtools merge. Then, peak calling was done with SICER with the following parameters: window size: 200; FDR controlling significance: 1e-2. Gap size parameters were determined according to the score value estimated by SICER: selected values of gap size are 600, 400, 800, and 1200 for H3K18ac, H3K9ac H3K27ac and H3K27me3, respectively. Differential enrichment analysis performed using SICER, similar to ChIPseq analysis.

## Data analysis and visualization

Browser views of gene tracks, ChIP-seq data and peaks were shown using Integrated Genomics Viewer (IGV; broadinstitute.org/igv). PCA, volcano plots, correlative heatmaps, boxplots, venn diagrams and scatter plots were generated using R for global comparison of samples and replicate analyses. Clustering analysis were performed with seqMINER v1.3.3g[70] using Refseq genes of mouse mm10 genome as reference coordinates or using Galaxy (Version 3.5.1.0.0). Metaprofiles were generated using Galaxy[71]. Reference regions were used as defined in results, blacklisted regions were removed, and signals were centered on Transcription start site or gene bodies. Functional enrichment analyses were performed with EnrichR (https://maayanlab.cloud/Enrichr/) using GO Biological Process 2023, ChEA 2022, and HDsigDB Mouse 2021 databases, and visualized with ggplot2 v3.5.0 using R[72]. Euchromatin score was computed using DESeq2 analysis. H3K27ac and H3K27me3 normalized read count per gene was calculated, as described[9], and z score values were calculated for each gene. Euchromatin score per gene corresponds to H3K27ac – H3K27me3 z score values. Co-expression analysis using co-expression modules identification tool (CEMiTool)[73] (https://cemitool.sysbio.tools/) was performed using euchromatin score calculated subtracting H3K27me3 to H3K27ac normalized read count per gene values (variance filter (*P*-value): 0.2; Variance stabilizing transformation: ON). Integrations were performed with published and accessible data in gene expression omnibus database: GSE65774[10], GSE152058[11], GSE227729[17], GSE157099[74].

**RNAseq analysis.** RNAseq datasets generated in the striatal bulk tissues and cells (dSPN, iSPN, astroglia) of HD KI mice and R6/2 (GSE65774[10], GSE152058[11]) were re-analyzed as described[74] starting from fastq files. STAR v2.7.10b was used to map reads and count the number of *reads* per gene using mm10 genome assembly[75] and Ensembl GRCm38 release 102[76]. Read counts were normalized across libraries with the method proposed by Anders and Huber[68]. The method implemented in DESeq2 was used to identify significantly differentially expressed genes between different mouse genotypes[69]. Resulting *P*-values were adjusted for multiple testing by using the Benjamini and Hochberg method.

## Protein expression analyses

One striatum per mouse was lysed and homogenized by pipetting in Laemmli buffer (BioRad), containing β-mercaptoethanol (1/40 of total volume). Samples were sonicated for $2 \times 10$ s (Bioblock Scientific Vibra Cell 75041, power 30%), heated 10 min at 70 °C, 5 min at 100 °C, centrifuged (15,000 × *g*, 5 min), and supernatant frozen at − 20 °C. Protein concentration was measured using the Qubit Protein Assay Kit (ThermoFisher). Protein samples were diluted for achieving equal protein concentrations and electrophoretic protein separation was carried out using 4–20% polyacrylamide gels (Criterion 500, BioRad) in TG-SDS buffer (Euromedex). Proteins were blotted onto nitrocellulose (Midi-Size Nitrocellulose, TransBlot Turbo, BioRad) using the TransBlot Turbo Transfer System (BioRad). Blots were blocked in 1% milk powder, then antibody against CBX4 (1:1000, Cell signaling - 30559), CBX6 (1:500, Cliniscience – SC-393040), CBX8 (1:1000, Cell signaling - 14696), Ezh1 (1:1000, LS bio – B13973), Ezh2 (1:1000, Diagenode – C15410039), Suz12 (1:1000, Cell signaling – 3737S) and Glis1 (1:1000, ThermoFisher, 23138-1-AP) were incubated overnight at 4 °C with 1% milk powder in washing buffer (Tris pH 7,4, NaCl 5 M, Tween 20%, distillated water). Blots were washed in washing buffer and secondary antibodies were added (horseradish peroxidase-conjugated whole-goat anti-rabbit/anti-mouse IgG (Jackson Laboratories). After 1 h incubation at room temperature and washes, blots were revealed with ECL (Clarity Western ECL Substrate, BioRad) and exposed with ChemiDoc Touch Imaging System (BioRad). The results were quantified using the ImageLab software (BioRad). All the present quantifications are relative to total protein levels.

## Immunohistological analysis

Mice were deeply anesthetized with ketamine/xylazine (100 mg/kg ketamine, 30 mg/kg xylazine) and intracardially perfused with saline (5 min) and 4% paraformaldehyde (10 min). Brains were extracted then stored in 20% sucrose in phosphate buffer during 48 h before freezing them in isopentane. Floating coronal sections (30 μm) were cut using a cryostat (CM3050S, Leica) in serial sections within a block of tissue extending from +1.34 to +0.02 from Bregma for the dorsal striatum. Sections were stored at –20 °C in tissue cryoprotective solution (30% glycerol, 30% ethyleneglycol, 40% phosphate buffer 0.1 M). Upon use, the sections were washed three times with PBS and incubated with blocking buffer (0.1% Triton-X-100 and 5% goat serum in PBS) for 2 h at room temperature, followed by incubation with primary antibodies in 0,1% Triton X-100 in PBS overnight at 4 °C: rabbit anti-CBX8 (1:500, #PA5-109483, Invitrogen) and mouse anti-NeuN (1:500, MAB377, Millipore), or rat anti-CTIP2 (1:250, ab123449, Abcam) and rabbit anti-NeuN (1:500, #ABN78, Millipore). Sections were then incubated with secondary antibody in 0,1% Triton X-100 in PBS for 2 h at room temperature before incubation with DAPI to stain nuclei. Secondary antibody used were: Alexa Fluor 488 AffiniPure anti-mouse (#A32723, Invitrogen) and Alexa Fluor 594 AffiniPure anti-rabbit (#A32754, Invitrogen), or Alexa Fluor 594 goat anti-rat (# A11007, Invitrogen) and Alexa Fluor 488 goat anti-rabbit (# A11008, Invitrogen). The images were acquired using a microscope Apotome (Apotome.2 Zeiss) using x20 (for CBX8-NeuN analysis) or x40 (for CTIP2-NeuN analysis) objectives.

Images for CBX8-NeuN quantification were analyzed using Zen Blue (Zeiss). Counting was performed manually, and the data were blinded to the assessor to avoid any bias. For CTIP2-NeuN quantification, six representative slices were analyzed per animal, with two pictures per region (dorsal striatum) per slice, images were analyzed using QuPath software. After a same threshold on DAPI channel using the "cell detection" tool of QuPath, an outline of each nucleus was obtained. A multiplex analysis was then performed to determine a minimal threshold of fluorescence intensity for each NeuN and CTIP2 channel to analyse, for each DAPI-positive nucleus, whether it was NeuN-positive, NeuN- and CTIP2-positive or negative for the both channels. Where multiple sections from one sample were assessed, the average of the counts per animal was used. For better visual representation, the fluorescence background was subtracted from 50 and a median filter of 0.1 was applied to the red and green channels (CBX8

and NeuN) using FIJI software. These corrections were not applied to image quantification.

## Statistics

For bar plots, centered regions indicate the mean +/-sem, for boxplots, centered regions indicate the median, box limits, upper and lower quartiles and whiskers, 1.5× interquartile range. All measurements were taken from distinct samples. For pairwise comparisons of average, data were tested for normality using the Shapiro's test. Statistical analyses included one-sample or two-tailed unpaired Student's $t$-test, two-way analysis of variance and two-ways analysis of variance with repeated measures. In case the samples were significantly non-normal, non-parametric tests, including Mann–Whitney, Kruskal–Wallis, one-sample Wilcoxon and binomial tests were performed. For multiple comparisons, the Bonferroni correction was applied. $P$ values < 0.05 were considered to be statistically significant, except when otherwise indicated. No statistical method was used to predetermine sample size, but our sample sizes are based on similar, previously established, experimental designs.

## Reporting summary

Further information on research design is available in the Nature Portfolio Reporting Summary linked to this article.

## Data availability

Source data are provided with this paper. Epigenomic data generated in this study have been deposited in the GEO database under accession codes GSE262075 [https://www.ncbi.nlm.nih.gov/geo/query/acc.cgi?acc=GSE262075] (FANS-ChIPseq R6/1 mice), GSE262076 [https://www.ncbi.nlm.nih.gov/geo/query/acc.cgi?acc=GSE262076] and GSE282238 [https://www.ncbi.nlm.nih.gov/geo/query/acc.cgi?acc=GSE282238] (FANS-CUT&Tag R6/1 mice), GSE262077 [https://www.ncbi.nlm.nih.gov/geo/query/acc.cgi?acc=GSE262077] (FANS-CUT&Tag temporal analysis HD Q140 KI mice) and GSE262078 [https://www.ncbi.nlm.nih.gov/geo/query/acc.cgi?acc=GSE262078] (ChIPseq R6/1 mice). RNAseq data used in this study are available in the GEO database under accession codes GSE65774 [https://www.ncbi.nlm.nih.gov/geo/query/acc.cgi?acc=GSE65774], GSE152058 [https://www.ncbi.nlm.nih.gov/geo/query/acc.cgi?acc=GSE152058], GSE227729 [https://www.ncbi.nlm.nih.gov/geo/query/acc.cgi?acc=GSE227729], GSE157099 [https://www.ncbi.nlm.nih.gov/geo/query/acc.cgi?acc=GSE157099]. Proteomic data used in this study are available at https://www.ebi.ac.uk/pride/archive/projects/PXD013771. Source data are provided with this paper.

## Code availability

Custom codes are available from the corresponding author upon request.

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

## Acknowledgements

We thank O. Bildstein, D. Egesi, G. Edomwonyi and A. Isik (LNCA UMR7364) for assistance in animal care. Sequencing was performed by the GenomEast Platform, a member of the 'France Génomique' consortium (ANR-10-INSN-0009), nuclei sorting was performed by the Flow Cytometry Platform at IGBMC. This study was supported by the Agence Nationale de la Recherche (ANR-2017-CE12-0027 and ANR-2022-CE12-0033 to K.M), the Fondation de la Recherche Medicale (FRM; Duban price to K.M.), the Centre National de la Recherche Scientifique (CNRS), the University of Strasbourg and the Interdisciplinary Thematic Institute NeuroStra (Strasbourg Uni, to KM and EB, as part of the ITI 2021-2028 program of the University of Strasbourg, CNRS and Inserm, supported by IdEx Unistra (ANR-10-IDEX-0002) and under the framework of the French Program "Investments for the Future"). R.A.V. was supported by post-doctoral fellowship from the IdEx fellowship program (Strasbourg Uni). J.Se. was supported by postdoctoral fellowship from the ANR (ANR-2017-CE12-0027). C.M. was supported by European postdoctoral fellowship (JPND programme, ANR-22-JPWG-0002-01). B.B. and J.Sc were recipients of doctoral fellowship from the French government. B.B. received PhD extension funded by NeuroStra (Strasbourg Uni). N.P. was supported by doctoral fellowship from the ANR (ANR-2022-CE12-0033).

## Author contributions

B.B. and R.A.V. performed epigenomic experiments, analysed the data and wrote the manuscript; N.P. performed epigenomic, immunohistological and western-blotting experiments, analysed the data and wrote the manuscript; J.Sc., C.M., and B.C. performed immunohistological data; J.Se and S.V.K. analysed epigenomic data; B.B. collected mouse tissues; S.L., C.D., and E.B. analysed epigenomic data and wrote the manuscript; A.L.B discussed the data and wrote the manuscript; K.M. designed and coordinated experiments, analysed the data, wrote the manuscript and secured funding.

## Competing interests

The authors declare no competing interests.
