## [Transparent Peer Review file · Nature Communications]

Accelerated epigenetic aging in Huntington's disease involves polycomb repressive complex 1

Corresponding Author: Dr Karine Merienne

Version 0:

Reviewer comments:

Reviewer #1

(Remarks to the Author)

The authors have used FANs with CUT&Tag seq to interrogate the striatal differential epigenetic status of dysregulated genes in Huntington's disease (HD). They perform this investigative analysis using striata from the R6/1 mouse model of HD and their wild-type counterparts and then use existing RNA-seq datasets from R6/2, knock-in mice, and in one case, human post-mortem brain, to provide independent data to support their findings. They found that neurodevelopmental genes are derepressed in HD striatal neurons, and that this was associated with general histone re-acetylation, a decrease in H3K27 trimethylation marks and H2K119 ubiquitylation. They demonstrated that this was catalyzed by polycomb repressive complexes 1 and 2 and that a PRC1-dependent subcluster of bivalent developmental transcription factors was more selectively re-activated in HD striatal neurons. Analysis of an enhancer in the top euchromatinized region in R6/1 as compared to wild-type striatal neurons suggested that the mechanism involved progressive paralog switching between PRC1-CBX genes. Loss of epigenetic information during physiological aging triggers the derepression of developmental genes and compromises cellular identity. Therefore, they analysed the epigenetic profiles of striatal genes in Q140 knock-in mice at 2, 6 and 10 months of age, and demonstrated that this aging signature is accelerated in mouse models of HD.

Transcriptional dysregulation is a well-defined pathogenic mechanism in HD and therefore, this mechanistic study will be of interest to wide base of researchers working on HD and other triplet repeat diseases. In this rigorous study, the authors have performed an extremely detailed and comprehensive analysis of the molecular basis and consequences of dysregulated transcriptional profiles in the striatum of Huntington's disease mice. They propose a number of mechanisms through which the depression of the polychrome complexes might have occurred, which will form the basis of future studies.

There are a few very minor comments.

Figure 1 and results

State the age of the R6/1 mice in the text and legend

Page 4 – top paragraph: typo dPSN instead of dSPN.

Figure 4e – very difficult to see the immunofluorescence. The panels could be made larger as there is ample white space in the figure. A higher-powered insert could be included in each panel.

Reviewer #2

(Remarks to the Author)

The study by Brule et al. advances our understanding of striatal epigenetics in Huntington's disease (HD) pathogenesis. They observed de-repression of neurodevelopmental genes and potential silencing of cell identity genes in neuronal nuclei of the striatum, which did not occur in non-neuronal nuclei. These changes were partly associated with the activation of bivalent promoters. Mechanistically they showed this likely occurs due to a paralogue switch in CBX subunits of the PRC1 complex to a configuration that is observed during neurodevelopment. They further delineate the temporal epigenetic changes in wildtype and Huntington mice at neurodevelopmental and stress-response genes. They further explored the use of histone modifications as an epigenetic clock to predict neuronal age. Overall, the manuscript will be of interest to the fields

of ageing, neurodegeneration and neuroepigenomics. Addressing the following comments would improve data interpretability further.

Comments

1. A significant limitation is the sample size for several of the epigenetic experiments. It appears that apart from the Q140 KI mice ($n = 3$ per group), most of the epigenetic experiments had $n=2$ or lower. This is a small sample size for comparisons between groups with low statistical power with possible dispersion errors in the DEseq2 modelling for differential methylated and acetylated regions between HD and WT. Also, the sample number should be more clearly indicated in the figure legend for each of the experiments.
2. There is a reduction of H3K27ac at neuronal-specific enhancers and striatal neuron identity genes in HD compared to controls. (Figure 1b and extended 1b). Additionally, at bivalent promoters, there is a decrease in H3K27me3 and an increase in histone acetylation in the R6/1 model (Fig 3d). Are these differences significant? It would be helpful to include some statistical analysis to support whether these differences are relevant.
3. What subtypes of neurons are present in the striatum and how are these represented in the NeuN+ve data? Could some of the epigenetic differences observed in NeuN+ve nuclei be due to changes in neuronal subtype proportions (due to selective vulnerability) in the striatum?
4. Figure 1 and 2 were reported to show that the “HD gene leads to heterochromatinization of identity genes and euchromatinization of developmental genes in vulnerable neurons”. The analysis was mostly focused on de-repression of developmental genes. A deeper analysis of identity genes would strengthen the above statement.
5. It was unclear whether some of the data was generated using bulk striatal samples or sorted NeuN nuclei, such as Fig 3a and Fig 5a (H2AK119ub). The legend for Fig. 6h states that bulk tissue was used but does not appear to be stated in the results. If bulk tissue is used, this needs to be clearly communicated in the results. Furthermore, would changes in cell type proportion affect the results for data generated using bulk striatal tissue (in disease contexts)? And can changes in cell type proportion be estimated and accounted for?
6. Regarding figure 5e, the text states “Further, integration with transcriptomic data showed that cluster 1 genes were specifically up-regulated in iSPN of HD mice (Fig. 5e and Extended data Fig. 5f).” This appears to be misleading as compared to controls the genes are not significantly upregulated ($p=0.075$). This should be more accurately described.
7. Figure 6 showed temporal changes in H3K27me3 and H3K27ac in an HTT model and controls. The interpretation of the findings was focused on the more pronounced changes that occur in the HTT model, however, there are clear changes (in particular H3K27me3) in the controls with age. It would be helpful to know whether the ageing-associated changes in the control mice are at the same or distinct loci to the HTT model. This would help delineate whether the HTT phenotype is accelerated ageing (same genes) versus aberrant ageing (different genes).
8. Regarding the co-expression analysis using CEMiTools, is it possible to tease out an M2 module interaction network to find out how *Onecut1* interacts with the other bivalent CBX genes?
9. The interpretation of figure 6 is that there is a “stronger contribution of H3K27me3 than H3K27ac to epigenetic aging in HD striatal neurons (Fig. 6f), further supporting critical role for PcG proteins.” However, it appears that H3K27me3 changes occurred in both controls and HTT mice and that increases in H3K27ac only occurred in the HTT mice (eg. Fig. 6f). An alternative interpretation is that a lack of H3K27ac changes in controls is what prevents (or slows down) the aberrant ageing-associated changes in gene expression. Therefore, H3K27ac plays a potentially critical role in this process also and should be discussed.

Minor comments

1. How were cell type identity genes and super-enhancers defined for downstream analysis?
2. Page 6, the following sentence was a bit confusing: “H3K27me3 was also higher in cluster 1 than cluster 2, while histone acetylation levels were similar in both clusters (Fig. 5c). H2AK119ub was specifically reduced at cluster 1 promoters in R6/1 vs WT striatal samples, in contrast to H3K27me3, reduced to lesser extent in the two clusters (Fig. 5c,d and Extended data Fig. 5e).” Initially this was interpreted as H3K27me3 levels being higher in R6/1 vs WT, rather than the overall H3K27me3 levels being higher in cluster 1 compared to 2 (for both R6/1 and WT). The second sentence was misinterpreted as H3K27me3 not being reduced in cluster 1. While technically written correctly, perhaps they could be reworded so that the reader can more easily follow.
3. The following is also a bit confusing: “Developmental transcription factors in cluster 1 above identified also showed accelerated euchromatinization using euchromatin score (Fig. 6e).” The graph for Fig. 6e is titled “Neurodevelopmental genes (Cluster 1)”. What genes are included in this analysis, all cluster 1 genes or a subset of only the TFs?

Reviewer #3

(Remarks to the Author)

In this study Brule et al, investigate the epigenetic and transcriptional changes that occur in mouse models of Huntington neurodegenerative disease (HD). They perform transcriptomic, epigenetic and proteomic analysis from WT and HD primary isolated striatal neurons. They investigate epigenetic changes that occur in HD and explore the link between polycomb repressive complex 1 (PRC1) and premature epigenetic erosion and developmental gene mis-expression. They suggest that epigenetic erosion leads to ectopic silencing of identity genes and simultaneous loss of chromatin-based repression of developmental regulators. They say this epigenetic aging is accelerated in HD mouse neurons and that this results from, at least in part, PRC1 dysfunction that may arise due to altering PRC1 composition to a more progenitor like configuration.

The strength of this study is that the authors generate a large amount of data from multiple mouse models to address their research question and do so in primary isolated cell populations (striatal neurons). However poor or insufficiently replicated data, lack of statistical rigor, over-interpretation and concluding causality from correlation substantially weaken their findings.

I believe that many of the observations are likely real (observed epigenetic changes, PRC-subunit expression changes, misregulation of stress response genes etc.) however the conclusions should be substantially tempered as all of the results are correlative only and do not show a direct contribution of epigenetic mechanisms in the observed phenotypes. In conclusion, whilst the subject area is clearly important and of interest to those studying neurodegenerative disease, aging and chromatin-based mechanisms of gene regulation the data is not sufficiently robust for publication in Nature Communications. I would suggest revising the data and analysis and then describing the observed correlations for what they are and leaving the speculations for the discussion.

Main comments

In the models being investigated, some mechanism is leading to the production of more immature striatal neurons. However, as epigenetic changes and gene expression have a two-way relationship, all the observed changes are merely indicative of gene expression changes. This extends to changes in PcG gene expression which, even when this occurs at the protein level (e.g. CBXs) is not direct evidence that this is the mechanisms. Indeed, it is just as likely that a switch to a more immature/progenitor like state leads to wholesale changes in gene expression programme that will include changes in many developmental regulators including the observed changes in PcG subunits. I do not believe that the authors have demonstrated the title – ‘Accelerated epigenetic aging in Huntington’s disease involves polycomb repressive complex 1’.

Specific Comments

The whole bivalent section (Figure 3 and associated) is weak and largely uninformative. You identified bivalent TSSs based on histone modifications and then found that they are indeed enriched for bivalent terms and PcG binding. It is a circular argument. There are then changes in modification state the match gene expression.

There is a switch in PRC1 subunit expression (indeed the data including - Extended Data Fig. 4e is nice) however there is no evidence this changes the stoichiometry of the PRC1 complex (no MS or quantitative immunoblots of changes in PRC1 composition) or proof that the expression of these alters the activity or targeting of the PRC1 complex to bring about the observed changes. As it is, these are simply developmental marker expressions as with *Onecut* and *Pax6* and do not inform mechanism. As such the statement - ‘Collectively, those data indicate that PRC1-CBX paralog switch is implicated in developmental gene de-repression in HD striatal neurons.’ is not justified.

The ChIP does not appear to be spike calibrated and as such a global ‘all gene’ change in modification state is impossible to determine using conventional normalisation. Also, more candidate example of individual loci and individual replicates is required to determine the robustness of the genomics data throughout. This refers to the ChIP data (and Cut and Run) in general and in particular data shown in Figure S3B /S5A&B.

Page 3 – ‘These data show that epigenetic dysregulation is greater in neurons compared to non-neuronal cells (e.g. glial cells) in HD mouse striatum.’ It should also be noted that *Neun-* will be a more heterogeneous population which may obscure some signals due to a dilution effect.

Figure 3D – The data is weak here. The differential is small which may be real but it requires some statistical analysis. Is this replicated? You comment on the changes in the WT vs HD *Neun+* population but ignore equivalent but more variable changes in the *Neun-* population of similar or greater magnitude (where incidentally the different acetylations do not all show the same trend – which is possible but seems a little unlikely). Needs statistical analysis across replicates, or at the very least data from both replicates shown.

Figure 3E – Two sample z-score heatmaps are not very informative if replicates are not shown. Also the heatmap in Supp 3C is mis-aligned on the right hand side.

Fig 4E is poorly presented. It is too small, low resolution and colour-blind unfriendly (at least include each channel individually). If you quantified *dapi* or *NeuN* it would likely give a similar result to that plotted graphically. Also, the differential values in the quantified immunoblot look insufficient to explain the proposed difference in the IF.

Page 5 – ‘promoting stoichiometry observed in immature neurons’. Where was this shown (here or in a reference? This should be indicated).

If you cluster as you have done in Fig 5a for K27me3 and then plot these groups as in 5d – do you find that K27 changes are bigger than *ub* changes? I don’t believe that this statement (P6) – ‘H2AK119ub was specifically reduced at cluster 1 promoters in R6/1 vs WT striatal samples, in contrast to H3K27me3, reduced to lesser extent in the two clusters (Fig. 5c,d and Extended data Fig. 5e). Thus, impaired activity of PRC1 might drive de-repression of developmental genes in HD vulnerable neurons, promoting de-repression neurodevelopmental transcription factors in cluster 1 that control neuronal fate’.

Page 6/7. ‘Principal component analysis (PCA) analysis showed that H3K27ac and H3K27me3 samples clustered according to genotype and age (Extended Data Fig. 6a).’ Only very weakly.

Fig 6b. The raw data is noisy (is this a single replicate or sum/average of multiple)?

Fig. 6e is underpowered for this analysis. 3 points is not robust enough to provide an epigenetic clock from which defined lengths of relative temporal shift can be robustly determined. Epigenetic clocks require massive amounts of data on which to

model such trends.

Figure 6K. All modules as displayed have a difference compared to WT not just 2 and 4.

Page 8. 'This suggests that epigenetic mechanism triggers early stress response in HD striatal neurons'. There is no evidence for this. Gene expression has changed so, so has the chromatin landscape. Can't tell cause and effect.

Minor.

'Euchromatinized' is not a commonly used word for the observed change in chromatin landscape, this should be rephrased.

Fig S1C – scale profiles to the same y per IP so that true comparisons of signal and differential signal can be determined.

Page 7. 'stronger contribution of H3K27me3 than H3K27ac to epigenetic aging in HD striatal neurons (Fig. 6f)' – This is correlative.

Fig 7C and D. Y axis values are too small to discern

I did not have access to Sup Tables.

Version 1:

Reviewer comments:

Reviewer #1

(Remarks to the Author)

The authors have satisfied the concerns of this referee

Reviewer #2

(Remarks to the Author)

Brule et al have made substantial revisions to their manuscript including multiple new datasets, which has significantly improved the power and confidence of their epigenomic findings. This has been followed by new analysis that included the contribution of healthy ageing versus Huntington's disease (HD) that highlighted an important role of H3K27ac in HD pathology. They have also made a number of additional improvements, including whether cell type proportion contributed to changes in their bulk analysis through immunostaining experiments of dSPN and iSPN neurons. Overall, the revisions have greatly improved the conclusions and interpretability of their study on the epigenomic contributions underlying HD.

Reviewer #3

(Remarks to the Author)

I have reviewed your manuscript that you have updated based on the three peer reviewer' comments. Whilst I still have some reservations about the cause vs. correlation aspects of the epigenetic data presented as outlined in my original review, I believe the changes you have made (textual, experiment and analytical) have substantially strengthened your arguments and the study in general. In addition, I admit that functionally addressing this relationship, particularly in this type of experimental model, is very challenging and beyond the scope of this study. I believe that the more extensive and toned down manuscript is now suitable and am happy to recommend it for publication in Nature Communications.

REVIEWER COMMENTS

Reviewer #1 (Remarks to the Author):

The authors have used FANs with CUT&Tag seq to interrogate the striatal differential epigenetic status of dysregulated genes in Huntington's disease (HD). They perform this investigative analysis using striata from the R6/1 mouse model of HD and their wild-type counterparts and then use existing RNA-seq datasets from R6/2, knock-in mice, and in one case, human post-mortem brain, to provide independent data to support their findings. They found that neurodevelopmental genes are derepressed in HD striatal neurons, and that this was associated with general histone re-acetylation, a decrease in H3K27 trimethylation marks and H2K119 ubiquitylation. They demonstrated that this was catalyzed by polycomb repressive complexes 1 and 2 and that a PRC1-dependent subcluster of bivalent developmental transcription factors was more selectively re-activated in HD striatal neurons. Analysis of an enhancer in the top euchromatinized region in R6/1 as compared to wild-type striatal neurons suggested that the mechanism involved progressive paralog switching between PRC1-CBX genes. Loss of epigenetic information during physiological aging triggers the derepression of developmental genes and compromises cellular identity. Therefore, they analysed the epigenetic profiles of striatal genes in Q140 knock-in mice at 2, 6 and 10 months of age, and demonstrated that this aging signature is accelerated in mouse models of HD.

Transcriptional dysregulation is a well-defined pathogenic mechanism in HD and therefore, this mechanistic study will be of interest to wide base of researchers working on HD and other triplet repeat diseases. In this rigorous study, the authors have performed an extremely detailed and comprehensive analysis of the molecular basis and consequences of dysregulated transcriptional profiles in the striatum of Huntington's disease mice. They propose a number of mechanisms through which the depression of the polychrome complexes might have occurred, which will form the basis of future studies.

There are a few very minor comments.

Figure 1 and results

State the age of the R6/1 mice in the text and legend

Page 4 – top paragraph: typo dPSN instead of dSPN.

Figure 4e – very difficult to see the immunofluorescence. The panels could be made larger as there is ample white space in the figure. A higher-powered insert could be included in each panel.

We thank the reviewer for the constructive comments and suggestions.

The age of R6/1 mice is now specified in the Results "...we generated ChIP-seq or CUT&Tag data on NeuN+ and NeuN- nuclei prepared from striatal tissue of 15 to 20 week-old symptomatic HD R6/1 transgenic and control mice (Fig. 1a and Supplementary Fig. 1a,b)" and in the Legend of Fig. 1 p19 "a. Scheme illustrating FANS-ChIPseq and FANS-CUT&Tag experiments conducted on 15-20 week-old R6/1 and WT mice."

The typo p.4 is corrected

Previous Fig. 4e (current Fig. 4f) was improved: the panels were enlarged, each channel is shown individually using colour-friendly selection, the resolution was improved and high-powered inserts were included in each panel.

Reviewer #2 (Remarks to the Author):

The study by Brule et al. advances our understanding of striatal epigenetics in Huntington's disease (HD) pathogenesis. They observed de-repression of neurodevelopmental genes and potential silencing of cell identity genes in neuronal nuclei of the striatum, which did not occur in non-neuronal nuclei. These changes were partly associated with the activation of bivalent promoters. Mechanistically they showed this likely occurs due to a paralogue switch in CBX subunits of the PRC1 complex to a configuration that is observed during neurodevelopment. They further delineate the temporal epigenetic changes in wildtype and Huntington mice at neurodevelopmental and stress-response genes. They further explored the use of histone modifications as an epigenetic clock to predict neuronal age. Overall, the manuscript will be of interest to the fields of ageing,

neurodegeneration and neuroepigenomics. Addressing the following comments would improve data interpretability further.

We thank the reviewer for the constructive comments and suggestions.

Comments

1. A significant limitation is the sample size for several of the epigenetic experiments. It appears that apart from the Q140 KI mice (n =3 per group), most of the epigenetic experiments had n=2 or lower. This is a small sample size for comparisons between groups with low statistical power with possible dispersion errors in the DEseq2 modelling for differential methylated and acetylated regions between HD and WT. Also, the sample number should be more clearly indicated in the figure legend for each of the experiments.

We have generated new FANS-CUT&Tag datasets in striatal neurons of R6/1 and control mice to increase biological replicates. Particularly, we have generated additional H3K9ac NeuN+ CUT&Tag data (n= 4-5 biological replicates, PCA in Fig. S2d), new H3K27me3 NeuN+ CUT&Tag data (n = 3 biological replicates, PCA in Fig. S5g) and new H2AK119ub NeuN+ CUT&Tag data (n = 3 biological replicates, PCA in Figure S5g). The new data have been used in the analyses shown Fig. 2f, g, in Fig. S2e (H3K9ac) and in Fig. S5h, i (H3K27me3 and H2AK119ub), they are described in the result section, p.6.

“H2AK119ub was specifically reduced at cluster 1 promoters in R6/1 vs WT striatal samples (Fig. 5b). In contrast, H3K27me3 depletion in R6/1 vs WT striatal neurons was similar in both clusters (Fig. 5d). Additionally, H3K27ac was similarly increased in both clusters in R6/1 striatal neurons (Fig. 5d). Kmeans clustering of bivalent genes using H3K27me3 data, instead of H2AK119ub data, supported the results, showing specific decrease of H2AK119ub in gene cluster normally strongly repressed (Supplementary Fig. 5f). Finally, to better compare H2AK119ub and H3K27me3 changes in R6/1 striatal neurons, we generated H2AK119ub and H3K27me3 FANS-CUT&Tag data using same NeuN+ nuclei (Supplementary Fig. 5g,h). Data analysis confirmed that H2AK119ub, in contrast to H3K27me3, was specifically reduced in cluster 1, in R6/1 vs WT striatal neurons (Supplementary Fig. 5i).”

These additional analyses strengthen our conclusions. Particularly, new H2AK119ub CUT&Tag data were generated in striatal neurons, while initial H2AK119ub ChIPseq data were generated in bulk striatal tissue. Thus, new H2AK119ub data show that H2AK119ub changes at cluster1 occurs in neurons. We did not generate new H3K27ac data, since our previous epigenomic analyses generated using bulk striatal tissue of HD-R6/1 and HD-Q140 mice showed loss of H3K27ac at striatal neuron identity genes (Achour et al. 2015 -ref 8-, Alcalá Vida et al. 2021 -ref 9-). H3K27ac data generated here, using sorted striatal neurons, confirm that these H3K27ac changes occur in neurons. Biological replicate numbers are specified in the legends of Fig. 1a, 5a and 6a, which illustrate experimental design of epigenomic experiments. Biological replicate numbers are also shown in PCA, Fig. S1a, S2d, S5g and S6a.

2. There is a reduction of H3K27ac at neuronal-specific enhancers and striatal neuron identity genes in HD compared to controls. (Figure 1b and extended 1b). Additionally, at bivalent promoters, there is a decrease in H3K27me3 and an increase in histone acetylation in the R6/1 model (Fig 3d). Are these differences significant? It would be helpful to include some statistical analysis to support whether these differences are relevant.

We have now quantified metaprofile data. For each analysis, quantifications represent variations of signals in HD vs WT. The quantifications were performed using the different biological replicates. Fig. 1c, S1d, 3b,c, S3d, 5b,d and S5f,i were changed accordingly. Statistical comparisons were performed between specific groups of genes/enhancers (e.g. neuronal identity genes vs all genes) or between same group of genes/enhancers in NeuN+ vs NeuN- samples.

The quantifications show that H3K27ac and H3K27me3 are significantly reduced and increased, respectively, at SPN identity genes (genes in M2 module, identified in WGCNA by Langfelder et al. Nat Neurosci. 2016, ref 10) and neuronal-specific enhancers (Fig. 1c and S1c), whereas the same marks are significantly increased and reduced at bivalent genes, respectively (Fig. 3b,c), which support our initial conclusions.

3. What subtypes of neurons are present in the striatum and how are these represented in the NeuN+ve data? Could some of the epigenetic differences observed in NeuN+ve nuclei be due to changes in neuronal subtype proportions (due to selective vulnerability) in the striatum?

dSPN (Drd1-enriched) and iSPN (Drd2-enriched) represent most vulnerable neuronal populations in HD. They also represent ≈95% of striatal neurons, remaining 5% being interneurons. To assess whether epigenetic

differences might be due to changes in neuronal sub-type proportions, we performed immunohistochemistry analysis using NeuN+ and Ctip2, a marker of both dSPN and iSPN (new Fig. S1e). The data show that numbers of Ctip2+ neurons and numbers of neurons are unchanged between R6/1 and WT striatum. The results are described p. 3

“...The effect appeared specific to neurons since non-neuronal-specific enhancers did not show loss of H3K27ac and gain of H3K27me3 in R6/1 vs WT NeuN- samples (Supplementary Fig. 1d). The numbers of neurons, including SPN (CTIP2+), were not different between R6/1 and WT mice, indicating that histone mark changes resulted neither from altered cell-type proportion or neuronal loss (Supplementary Fig. 1e).”

4. Figure 1 and 2 were reported to show that the “HD gene leads to heterochromatinization of identity genes and euchromatinization of developmental genes in vulnerable neurons”. The analysis was mostly focused on de-repression of developmental genes. A deeper analysis of identity genes would strengthen the above statement.

We focused on epigenetic changes at developmental genes, since in early studies using bulk striatal tissue of HD mice, we described loss of H3K27ac at neuronal identity genes (regulated by super-enhancers), which correlated with transcriptional down-regulation (Achour et al. HMG 2015 ref 8, Alcalá Vida et al. Nat Commun 2021 ref 9). This is mentioned in the introduction p.2:

“Epigenomic analyses in bulk HD striatal tissue show early loss of H3K27ac at striatal super-enhancers -broad enhancers regulating cellular identity genes, which results in transcriptional down-regulation of their targets, striatal identity genes⁸⁻¹². However, due to lack of cellular and temporal resolutions of earlier analyses, it was unclear whether HD vulnerable neurons undergo accelerated epigenetic aging, including accelerated de-repression of developmental genes”.

The data generated in the current study are more specific since they demonstrate that neurons are responsible for those changes. Importantly, they also show that the HD ‘identity’ signature associates with epigenetic de-repression of developmental genes, thus providing strong support to accelerated epigenetic erosion/aging in the pathogenic mechanism that affects HD vulnerable neurons. We make clearer the fact that SPN identity genes are epigenetically repressed in R6/1 vs WT striatal neurons, 1) quantifying H3K27ac loss and H3K27me3 gain at SPN identity genes and neuronal-specific enhancers (Fig. 1c and Fig. S1c) and 2) showing igv profiles of our data at SPN identity genes, including *Drd1* (dSPN) and *Drd2* (iSPN) (Fig. 1b and Fig. S1c). The text is modified accordingly p.3 :

“Consistent with earlier bulk H3K27ac ChIPseq analyses, in R6/1 mice, H3K27ac signals in striatal neurons was reduced at neuronal-specific enhancers and SPN identity genes regulated by super-enhancers (e.g. *Drd2*, *Drd1*), which associated with increased H3K27me3 levels (Fig. 1b,c and Supplementary Fig. 1b). The effect appeared specific to neurons since non-neuronal-specific enhancers did not show loss of H3K27ac and gain of H3K27me3 in R6/1 vs WT NeuN- samples (Supplementary Fig. 1b).”

Additionally, recent snRNAseq study using striatal tissue of HD patients (Handsaker et al. BioRxiv 2024, ref 25) shows that SPN of HD patients undergo transcriptional erosion, which is characterized by both down-regulation of identity genes and de-repression of developmental genes. This new study that was not released at the time of submission of our article, it is now discussed, p9:

“Due to the scarcity of human data, particularly the lack of temporal epigenomic and transcriptomic data, it is difficult to extend the results to humans. Nonetheless, recent transcriptomic data suggest that the mechanism occurs in humans. First, using published FANS-seq data²⁴, we found that bivalent genes, including *ONECUT1*, *PAX6* and *RUNX2*, were up-regulated in SPN of HD patients. Second, simultaneous single cell analysis of CAG expansion and transcriptome shows that long CAG expansions that result from somatic instability induce deleterious transcriptional erosion in SPN of HD patients²⁵. This erosion is similar to that observed in HD mice: it is characterized by progressive down-regulation of SPN identity genes as well as de-repression of developmental genes normally repressed, so-called de-repression crisis.”

5. It was unclear whether some of the data was generated using bulk striatal samples or sorted NeuN nuclei, such as Fig 3a and Fig 5a (H2AK119ub). The legend for Fig. 6h states that bulk tissue was used but does not appear to be stated in the results. If bulk tissue is used, this needs to be clearly communicated in the results.

Furthermore, would changes in cell type proportion affect the results for data generated using bulk striatal tissue (in disease contexts)? And can changes in cell type proportion be estimated and accounted for?

The information is clearly specified in the legends of the figures. Note that, while initial H2AK119ub datasets were generated on bulk striatal tissue, new H2AK119ub datasets were generated on NeuN+ striatal tissue. To make it clearer, we have included a scheme representing H2AK119ub-related experiments (new Fig.5a). Importantly, analyses performed using NeuN+ H2AK119ub data support bulk H2AK119ub data. Particularly, they confirm that H2AK119ub is most reduced at cluster1 (Fig. S5i). Given that new experiment was performed on neuronal population and we show that there is no significant change in Ctip2 proportion in R6/1 vs WT striatum (shown in Fig. S1e), it is unlikely that changes in cell type proportions underlies our results. This is discussed p.3:

“SPN numbers were not different between R6/1 and WT mice, indicating that histone mark changes did not result from alteration in neuronal type proportions or neuronal loss (Supplementary Fig. 1e).”

6. Regarding figure 5e, the text states “Further, integration with transcriptomic data showed that cluster 1 genes were specifically up-regulated in iSPN of HD mice (Fig. 5e and Extended data Fig. 5f).” This appears to be misleading as compared to controls the genes are not significantly upregulated ($p=0.075$). This should be more accurately described.

The text has been changed accordingly: “Further, integration with transcriptomic data showed that cluster 1 genes specifically showed a trend to up-regulation in iSPN of HD mice (Fig. 5e and Supplementary Fig. 5f).”

7. Figure 6 showed temporal changes in H3K27me3 and H3K27ac in an HTT model and controls. The interpretation of the findings was focused on the more pronounced changes that occur in the HTT model, however, there are clear changes (in particular H3K27me3) in the controls with age. It would be helpful to know whether the ageing-associated changes in the control mice are at the same or distinct loci to the HTT model. This would help delineate whether the HTT phenotype is accelerated ageing (same genes) versus aberrant ageing (different genes).

We performed the analysis suggested by the reviewer (Fig. S6f,g). The results show age-dependent H3K27me3 and H3K27ac changes in WT samples. H3K27me3 was predominantly reduced with age in WT, while H3K27ac was predominantly increased with age in WT. HD- and age-mediated H3K27me3 depleted genes significantly overlapped. HD- and age-mediated H3K27ac enriched genes also significantly overlapped (Fig. S6g). Functional enrichment analysis showed that both H3K27me3-depleted genes and H3K27ac-enriched genes with age in WT have prominent developmental signatures (Fig. S6f), suggesting that H3K27me3 and H3K27ac both contribute to physiological age-dependent de-repression of developmental genes. Interestingly however, loss of H3K27me3 appeared greater than gain of H3K27ac in normal aging, in contrast to HD. Also, H3K27me3 and H3K27ac at PRC1-CBX locus/genes (i.e. Cbx4 & Cbx8) were depleted and enriched, respectively, in HD samples but not in WT samples during aging (Fig. S6g), supporting specific role for PRC1-CBX in HD.

This is described in the manuscript p.7: “Age-dependent variations of H3K27me3 and H3K27ac in cluster 1 genes suggest that H3K27me3 contributes to epigenetic age in both HD and WT striatal neurons, whereas H3K27ac may have specific role in HD (Fig. 6f). Differential analysis of H3K27me3 and H3K27ac levels in WT samples showed that genes depleted in H3K27me3 and increased in H3K27ac with age, both displayed developmental signatures, including signature related to “Nervous system development” (Supplementary Fig. 6f). Moreover, H3K27me3-depleted genes in HD vs WT and in old vs young samples significantly overlapped (Supplementary Fig. 6g). This was also the case when considering H3K27ac-increased genes (Supplementary Fig. 6g). This indicates that common mechanism contributes to de-repression of developmental genes in normal aging and in HD. Interestingly however, the analysis also suggests greater involvement of H3K27ac in HD and of H3K27me3 in normal aging, since changes in H3K27ac and in H3K27me are predominant in HD and normal aging, respectively (Supplementary Fig. 6g).”

Finally, although PRC1-CBX is implicated in transcriptional repression, recent studies show that Cbx4 can promote transcriptional up-regulation. Specifically, this was shown to occur at Runx2, a pioneer, developmental transcription factor involved in skeletal development (Wang et al. Nat Commun 2020 ref 44; Hojo et al. Cell Rep 2022 45). Cbx4 was found to lead to increased H3K27ac at Runx2 (Wang et al. Nat Commun 2020). Remarkably, Runx2 is within the gene list described in Fig.2c, which contains H3K27me3-depleted and H3K27ac-enriched genes in R6/1 vs WT striatal neurons. Significant functional interaction network was

performed using this gene list, which identified major network comprising Cbx4, Runx2, Pax6 and Onecut1 (Fig. 2i). Moreover, transcriptomic data show that Runx2 is up-regulated in HD SPNs -more so in iSPNs- and in age-dependent manner (Fig. S2g and S6e). H3K27ac and H3K27me3 temporal data further show accelerated epigenetic de-repression of Runx2 in Q140 striatal neurons (Fig. S6e). Thus, specific up-regulation of Cbx4 in HD vs WT striatal neurons may trigger increased histone acetylation and de-repression of Runx2, which as a pioneer factor, could drive de-repression of developmental transcription factors. While we feel testing the hypothesis is beyond the scope of this study, we discuss this intriguing possibility:

P11: “Additionally, PRC1-CBX changes might play more direct role in de-repression of developmental genes in HD striatal neurons. Strikingly, CBX4 can act as a transcriptional activator, which recruits the histone acetyltransferase GCN5 and promotes histone acetylation, including H3K27ac, notably at the promoter of *Runx2*, a transcription factor implicated in skeletal development⁴⁴⁻⁴⁶. Interestingly, our data show that *Runx2* undergoes accelerated epigenetic de-repression in HD striatal neurons, particularly at early stage of the pathology. Since RUNX2 is a pioneer transcription factor, this might initiate de-repression of self-regulating developmental transcription factors, which drives positive autoregulatory feedback loop leading to de-repression crisis. In fact, such feedback loop driving re-activation of bivalent genes, including Cbx4 and autoregulatory transcription factors such as *Onecut1*, was described in striatal neurons deficient in PRC2³⁴. Intriguingly, this associated with repression of striatal neuron identity genes³⁴. These results further highlight critical role for polycomb repressive proteins in epigenetic erosion of striatal neuron identity, though their precise interplay needs to be addressed in the future.”

8. Regarding the co-expression analysis using CEMiTools, is it possible to tease out an M2 module interaction network to find out how Onecut1 interacts with the other bivalent CBX genes?

We could not retrieve significant interaction network using the M2 module. However, as indicated in above answer, significant functional interaction network was found using H3K27me3-depleted and H3K27ac-enriched genes in R6/1 vs WT striatal neurons shown in Fig.2c, and major network shows interactions between Cbx4, Runx2, Pax6 and Onecut1 (Fig. 2i).

9. The interpretation of figure 6 is that there is a “stronger contribution of H3K27me3 than H3K27ac to epigenetic aging in HD striatal neurons (Fig. 6f), further supporting critical role for PcG proteins.” However, it appears that H3K27me3 changes occurred in both controls and HTT mice and that increases in H3K27ac only occurred in the HTT mice (eg. Fig. 6f). An alternative interpretation is that a lack of H3K27ac changes in controls is what prevents (or slows down) the aberrant ageing-associated changes in gene expression. Therefore, H3K27ac plays a potentially critical role in this process also and should be discussed.

In the light of temporal analysis in WT samples suggested by the reviewer, we do agree with this view. We indeed find that histone acetylation changes over time are limited in WT compared to histone acetylation changes in HD vs WT, and the opposite is true for H3K27me3 as explained above (point 7).

p.7: “Age-dependent variations of H3K27me3 and H3K27ac in cluster 1 genes suggest that H3K27me3 contributes to epigenetic age in both HD and WT striatal neurons, whereas H3K27ac may have specific role in HD (Fig. 6f). Differential analysis of H3K27me3 and H3K27ac levels in WT samples showed that genes depleted in H3K27me3 and increased in H3K27ac with age, both displayed developmental signatures, including signature related to “Nervous system development” (Supplementary Fig. 6f). Moreover, H3K27me3-depleted genes in HD vs WT and in old vs young samples significantly overlapped (Supplementary Fig. 6g). This was also the case when considering H3K27ac-increased genes (Supplementary Fig. 6g). This indicates that common mechanism contributes to de-repression of developmental genes in normal aging and in HD. Interestingly however, the analysis also suggests greater involvement of H3K27ac in HD and of H3K27me3 in normal aging, since changes in H3K27ac and in H3K27me are predominant in HD and normal aging, respectively (Supplementary Fig. 6g).”

We also discuss the possibility that histone acetylation changes underlie accelerated de-repression of developmental genes in HD vulnerable neurons, through a mechanism implicating Cbx4 and Runx2, as mentioned above

P11: “Additionally, PRC1-CBX changes might play more direct role in de-repression of developmental genes in HD striatal neurons. Strikingly, CBX4 can act as a transcriptional activator, which recruits the histone acetyltransferase GCN5 and promotes histone acetylation, including H3K27ac, notably at the

promoter of *Runx2*, a transcription factor implicated in skeletal development⁴⁴⁻⁴⁶. Interestingly, our data show that *Runx2* undergoes accelerated epigenetic de-repression in HD striatal neurons, particularly at early stage of the pathology. Since RUNX2 is a pioneer transcription factor, this might initiate de-repression of self-regulating developmental transcription factors, which drives positive autoregulatory feedback loop leading to de-repression crisis. In fact, such feedback loop driving re-activation of bivalent genes, including *Cbx4* and autoregulatory transcription factors such as *Onecut1*, was described in striatal neurons deficient in PRC2³⁴. Intriguingly, this associated with repression of striatal neuron identity genes³⁴. These results further highlight critical role for polycomb repressive proteins in epigenetic erosion of striatal neuron identity, though their precise interplay needs to be addressed in the future.”

Minor comments

1. How were cell type identity genes and super-enhancers defined for downstream analysis?

In initial analysis, we used neuronal identity genes and glial identity genes as defined in previous study (Alcala Vida et al. Nat Commun 2021 ref 9), where we generated H3K27ac and H3K27me3 ChIPseq data on NeuN+ and NeuN- striatal cells using WT mouse striatum for the purpose of integrative analyses. Those data were also used to define neuronal-specific and glial-specific enhancers.

We have modified the analysis for simplification. Specifically, to generate metaprofiles in Fig.1b, we now use gene module enriched in SPN identity genes, which was identified by WGCNA in the study by Langfelder et al. 2016 Nat. Neurosci. 2016 (ref 10). This WGCNA analysis is widely used. The SPN identity gene module corresponds to the M2 module in the Langfelder study. We also generated metaprofiles using all genes, for comparison (Fig. 1b). The manuscript was changed accordingly p3.

“Consistent with earlier bulk H3K27ac ChIPseq analyses, in R6/1 mice, H3K27ac signals in striatal neurons was reduced at neuronal-specific enhancers and SPN identity genes regulated by super-enhancers (e.g. *Drd2*, *Drd1*), which associated with increased H3K27me3 levels (Fig. 1b,c and Supplementary Fig. 1b).”

2. Page 6, the following sentence was a bit confusing: “H3K27me3 was also higher in cluster 1 than cluster 2, while histone acetylation levels were similar in both clusters (Fig. 5c). H2AK119ub was specifically reduced at cluster 1 promoters in R6/1 vs WT striatal samples, in contrast to H3K27me3, reduced to lesser extent in the two clusters (Fig. 5c,d and Extended data Fig. 5e).” Initially this was interpreted as H3K27me3 levels being higher in R6/1 vs WT, rather than the overall H3K27me3 levels being higher in cluster 1 compared to 2 (for both R6/1 and WT). The second sentence was misinterpreted as H3K27me3 not being reduced in cluster 1. While technically written correctly, perhaps they could be reworded so that the reader can more easily follow.

We have rephrased the sentences as well as included new analyses:

P6: “H2AK119ub was specifically reduced at cluster 1 promoters in R6/1 vs WT striatal samples (Fig. 5b). In contrast, H3K27me3 depletion in R6/1 vs WT striatal neurons was similar in both clusters (Fig. 5d). Additionally, H3K27ac was similarly increased in both clusters in R6/1 striatal neurons (Fig. 5d). Kmeans clustering of bivalent genes using H3K27me3 data, instead of H2AK119ub data, supported the results, showing specific decrease of H2AK119ub in gene cluster normally strongly repressed (Supplementary Fig. 5f). Finally, to better compare H2AK119ub and H3K27me3 changes in R6/1 striatal neurons, we generated H2AK119ub and H3K27me3 FANS-CUT&Tag data using same NeuN+ nuclei (Supplementary Fig. 5g,h). Data analysis confirmed that H2AK119ub, in contrast to H3K27me3, was specifically reduced in cluster 1, in R6/1 vs WT striatal neurons (Supplementary Fig. 5i).”

3. The following is also a bit confusing: “Developmental transcription factors in cluster 1 above identified also showed accelerated euchromatinization using euchromatin score (Fig. 6e).” The graph for Fig. 6e is titled “Neurodevelopmental genes (Cluster 1)”. What genes are included in this analysis, all cluster 1 genes or a subset of only the TFs?

The mistake is corrected, the title of the graph in Fig.6e is “developmental TF (cluster 1)”.

Reviewer #3 (Remarks to the Author):

In this study Brule et al, investigate the epigenetic and transcriptional changes that occur in mouse models of Huntington neurodegenerative disease (HD). They perform transcriptomic, epigenetic and proteomic analysis from WT and HD primary isolated striatal neurons. They investigate epigenetic changes that occur in HD and explore the link between polycomb repressive complex 1 (PRC1) and premature epigenetic erosion and developmental gene mis-expression. They suggest that epigenetic erosion leads to ectopic silencing of identity genes and simultaneous loss of chromatin-based repression of developmental regulators. They say this epigenetic aging is accelerated in HD mouse neurons and that this results from, at least in part, PRC1 dysfunction that may arise due to altering PRC1 composition to a more progenitor like configuration.

The strength of this study is that the authors generate a large amount of data from multiple mouse models to address their research question and do so in primary isolated cell populations (striatal neurons). However poor or insufficiently replicated data, lack of statistical rigor, over-interpretation and concluding causality from correlation substantially weaken their findings.

I believe that many of the observations are likely real (observed epigenetic changes, PRC-subunit expression changes, misregulation of stress response genes etc.) however the conclusions should be substantially tempered as all of the results are correlative only and do not show a direct contribution of epigenetic mechanisms in the observed phenotypes. In conclusion, whilst the subject area is clearly important and of interest to those studying neurodegenerative disease, aging and chromatin-based mechanisms of gene regulation the data is not sufficiently robust for publication in Nature Communications. I would suggest revising the data and analysis and then describing the observed correlations for what they are and leaving the speculations for the discussion.

We thank the reviewer for the constructive comments and suggestions. As noticed by the reviewer, we generated large amount of epigenomic data, including FANS-ChIP-seq and FANS-CUT&Tag data using brain tissue. Those approaches remain technically challenging, particularly since they require large amounts of material (nuclei) not to mention the cost. CUT&Tag, which is recent, is actually interesting since it is less costly than ChIP-seq. However, making it work using sorted nuclei from frozen brain tissues has required optimization. In fact, we successfully set up FANS coupled to CUT&Tag using frozen brain tissue. The data we generated, whether ChIPseq or CUT&Tag, are of high quality. They can be visualized in GEO. Moreover, we have included new igv captures showing specific genomic regions and including biological replicates (Fig. 1c, Fig. S1d, Fig. S5b Fig. 6b, Fig. S6h). Finally, to improve robustness of our results, we have generated new H3K9ac, H3K27me3 and H2AK119ub CUT&Tag data using NeuN+ nuclei of R6/1 and WT mice (N=3-5 biological replicates). Those new datasets strengthen initial observations, which is detailed in answers provided below. Additionally, we have performed additional quantifications and statistical analyses that support our conclusions. Specifically, we have quantified changes in histone modifications levels at gene clusters of interest (Fig. 1b, S1c, 3b,c, S3d, 5d, S5f,i). Finally, we went carefully through the manuscript to toned-down when necessary. Overall, we believe our manuscript was substantially improved.

Main comments

In the models being investigated, some mechanism is leading to the production of more immature striatal neurons. However, as epigenetic changes and gene expression have a two-way relationship, all the observed changes are merely indicative of gene expression changes. This extends to changes in PcG gene expression which, even when this occurs at the protein level (e.g. CBXs) is not direct evidence that this is the mechanisms. Indeed, it is just as likely that a switch to a more immature/progenitor like state leads to wholesale changes in gene expression programme that will include changes in many developmental regulators including the observed changes in PcG subunits. I do not believe that the authors have demonstrated the title – ‘Accelerated epigenetic aging in Huntington’s disease involves polycomb repressive complex 1’.

We agree that epigenetic and transcriptional changes tightly interweave. However, we think strong arguments support the hypothesis that altered PRC plays a key role in de-repression of developmental genes in HD striatal neurons. First, deeper analysis of our data shows that de-repressed bivalent PRC target genes include pioneer transcription factors, particularly Runx2, which is a target of Cbx4 (Hojo et al. Cell Rep 2022 ref 45; Wang et al. Nat Commun 2020 ref 44). Interestingly, Cbx4 transcriptional up-regulation of Runx2 that was mediated by increased histone acetylation (H3K27ac; Wang et al. Nat Commun 2020). Thus, de-repression of Runx2, which is accelerated in HD mouse striatal neurons, particularly at early pathological stage (Fig. S6e), could trigger CBX4-dependent chromatin opening at Runx2 target genes. Second, de-repressed bivalent PRC target genes,

particularly genes in cluster 1, contain self-regulating transcription factors (e.g. *Onecut1*), which can be rapidly up-regulated through mutual reinforcement of transcriptional networks. In fact, such a mechanism was described in the context of PRC2 deficiency in mouse striatal neurons (von Schimmelmann et al. Nat Neurosci. 2016 ref 34). In this paper, the authors show that inactivation of *Ezh1/2* in striatal neurons leads to “de-repression of selected, predominantly bivalent PRC2 target genes that are dominated by self-regulating transcription factors normally suppressed in these neurons”. Consistently, network analysis of genes that are both depleted in H3K27me3 and enriched in H3K27ac in striatal neurons of R6/1 mice identifies a major functional interaction network that contains *Cbx4*, *Runx2*, *Pax6* and *Onecut1* (Fig. 2i). This is discussed p.11:

P11: “Additionally, PRC1-CBX changes might play a more direct role in de-repression of developmental genes in HD striatal neurons. Strikingly, CBX4 can act as a transcriptional activator, which recruits the histone acetyltransferase GCN5 and promotes histone acetylation, including H3K27ac, notably at the promoter of *Runx2*, a transcription factor implicated in skeletal development⁴⁴⁻⁴⁶. Interestingly, our data show that *Runx2* undergoes accelerated epigenetic de-repression in HD striatal neurons, particularly at early stage of the pathology. Since RUNX2 is a pioneer transcription factor, this might initiate de-repression of self-regulating developmental transcription factors, which drives a positive autoregulatory feedback loop leading to de-repression crisis. In fact, such a feedback loop driving re-activation of bivalent genes, including *Cbx4* and autoregulatory transcription factors such as *Onecut1*, was described in striatal neurons deficient in PRC2³⁴. Intriguingly, this is associated with repression of striatal neuron identity genes³⁴. These results further highlight a critical role for polycomb repressive proteins in epigenetic erosion of striatal neuron identity, though their precise interplay needs to be addressed in the future.”

Specific Comments

The whole bivalent section (Figure 3 and associated) is weak and largely uninformative. You identified bivalent TSSs based on histone modifications and then found that they are indeed enriched for bivalent terms and PcG binding. It is a circular argument. There are then changes in modification state that match gene expression.

We indeed identified bivalent TSSs based on striatal H3K4me3 and H3K27me3 ChIPseq data and kmeans clustering. We then verified, through functional enrichment analysis, that the identified bivalent cluster showed expected features, using functional enrichment analyses: GO BP, ChEA and HDSigDB analyses showed that the bivalent cluster was strongly enriched in developmental genes and PRC1/2 target genes. Those analyses have been moved from Fig. 3 to Fig. S3a,b,c. HDSigDB analysis (new Fig. S3c) also showed significant overlap between genes in our bivalent cluster and those that are bivalent and up-regulated in striatal neurons of PRC2 deficient mice in the study by von Schimmelmann et al. Nat Neurosci. 2016 (ref 34). Having checked that our bivalent cluster displayed expected features, we then assessed histone modification changes in bivalent cluster in R6/1 vs WT striatal neurons. We have improved those analyses, providing quantifications (Fig. 3b,c, Fig. S3d). We have also modified heatmaps of expression values of bivalent genes in iSPN and dSPN of HD mice and patients, representing log2FC instead of z-score values (Fig. 3d, f).

There is a switch in PRC1 subunit expression (indeed the data including - Extended Data Fig. 4e is nice) however there is no evidence this changes the stoichiometry of the PRC1 complex (no MS or quantitative immunoblots of changes in PRC1 composition) or proof that the expression of these alters the activity or targeting of the PRC1 complex to bring about the observed changes. As it is, these are simply developmental marker expressions as with *Onecut* and *Pax6* and do not inform mechanism. As such the statement - ‘Collectively, those data indicate that PRC1-CBX analog switch is implicated in developmental gene de-repression in HD striatal neurons.’ is not justified.

We include additional analysis (Fig. 4e) using temporal proteomic data generated by LC MS/MS with striatal tissue of HD R6/2 control animals of 1, 2 and 3 weeks of age (Project PXD013771 by Schaab C., deposited on PRIDE). *Cbx4*, *Cbx6*, *Cbx7* and *Cbx8* protein levels were detected in these data. The data show that *Cbx4* and *Cbx8* progressively increase over time, in contrast to *Cbx6* and *Cbx7*, showing reduced levels.

P5: “Finally, temporal transcriptomic data generated on striatal tissue of HD knockin (KI) mice, including Q140 and Q175 lines¹⁰, and proteomic data generated on the striatum of the HD-R6/2 line

both showed age-dependent increase of *Cbx4* and *Cbx8* and reduction of *Cbx6* and *Cbx7* in HD mice, suggesting progressive PRC1-CBX paralog switch (Fig. 4e and Extended Data Fig. 4e).”

Together with above precisions (*Cbx4* is a direct activator of *Runx2*; *Cbx4*, *Runx2*, *Pax6*, *Onecut1* are part of same functional network; up-regulated developmental transcription factors such as *Onecut1* show self-regulating properties), we believe these new results strengthen our conclusion. Nonetheless, we toned down the conclusion p.6 “Thus, PRC1-CBX paralog switch might be implicated in developmental gene de-repression in HD striatal neurons.”

The ChIP does not appear to be spike calibrated and as such a global ‘all gene’ change in modification state is impossible to determine using conventional normalisation. Also, more candidate example of individual loci and individual replicates is required to determine the robustness of the genomics data throughout. This refers to the ChIP data (and Cut and Run) in general and in particular data shown in Figure S3B /S5A&B.

We did not spike-in our ChIP-seq and CUT&Tag experiments, the reason why we did not conclude about global changes. Our conclusions are based on either differential analyses or comparisons between genomic regions/genes of interest versus all regions/genes.

We include additional loci and individual replicates in igv captures (Fig.1b and S1c, S5b,h). Moreover, visualization files (bigwig) have been deposited on GEO.

Page 3 – ‘These data show that epigenetic dysregulation is greater in neurons compared to non-neuronal cells (e.g. glial cells) in HD mouse striatum.’ It should also be noted that NeuN- will be a more heterogeneous population which may obscure some signals due to a dilution effect.

We include this precision p.3: “These data indicate that epigenetic dysregulation is dramatic in neurons compared and minimal in non-neuronal cells in HD mouse striatum. However, due to heterogeneity of NeuN- nuclei, we cannot fully rule out that specific non-neuronal cells (e.g. glial cells) undergo substantial epigenetic changes.”

Moreover, we now focus essentially on NeuN+ data (H3K9ac and H3K18ac data in NeuN- striatal cells were removed).

Figure 3D – The data is weak here. The differential is small which may be real but it requires some statistical analysis. Is this replicated? You comment on the changes in the WT vs HD NeuN+ population but ignore equivalent but more variable changes in the NeuN- population of similar or greater magnitude (where incidentally the different acetylations do not all show the same trend – which is possible but seems a little unlikely). Needs statistical analysis across replicates, or at the very least data from both replicates shown.

The changes were quantified using the different replicates. This is shown in new Fig. 3b,c. Specifically, H3K27ac and H3K27me3 changes at bivalent promoters were compared 1) between NeuN+ and NeuN- samples and 2) with all promoters. The results show that H3K27me3 is reduced in R6/1 NeuN+ vs NeuN- bivalent promoters as well as in R6/1 bivalent vs all promoters in NeuN+ samples, while changes in opposite direction are observed for H3K27ac. Fig. S3d further shows that H3K9ac and H3K18ac are increased at bivalent vs all promoters in NeuN+ samples.

Figure 3E – Two sample z-score heatmaps are not very informative if replicates are not shown. Also the heatmap in Supp 3C is mis-aligned on the right hand side.

We have removed those heatmaps and generated new heatmaps using Log2FC values (Fig. 3d,f). In Fig. 3f, we have used published transcriptomic data generated on dSPN and iSPN of HD patients (Matlik et al. Nat Genet 2024 ref 24). Overall, the data show that bivalent genes undergo prominent transcriptional up-regulation in SPN expressing the HD gene.

Fig 4E is poorly presented. It is too small, low resolution and colour-blind unfriendly (at least include each channel individually). If you quantified dapi or NeuN it would likely give a similar result to that plotted graphically. Also, the differential values in the quantified immunoblot look insufficient to explain the proposed difference in the IF.

The was improved (new Fig.4f): it was enlarged and colour-blind friendly, the resolution was also improved. The signal was quantified on NeuN+ cells only, whereas western-blotting analyses were performed on bulk striatal tissue. Moreover, we performed additional immunohistofluorescence experiments that show that Dapi and NeuN are not changed in R6/1 vs WT striatum (Fig. S1e).

Page 5 – ‘promoting stoichiometry observed in immature neurons’. Where was this shown (here or in a reference? This should be indicated).

The statement referred to above sentence: “*Cbx2*, *Cbx4* and *Cbx8* are substantially expressed in differentiating cells, including neural progenitor cells (NPC), and become down-regulated in differentiated cells¹⁹. Consistently, *Cbx2*, *Cbx4* and *Cbx8* expression was low in mature striatal neurons (Fig. 4b,c and Extended Data Fig. 4c).”

For clarity, we recall ref 19: “Together, this indicates that PRC1-CBX genes undergo paralog switch in HD SPN, promoting stoichiometry observed in immature neurons^{19”}.

If you cluster as you have done in Fig 5a for K27me3 and then plot these groups as in 5d – do you find that K27 changes are bigger than ub changes? I don’t believe that this statement (P6) – ‘H2AK119ub was specifically reduced at cluster 1 promoters in R6/1 vs WT striatal samples, in contrast to H3K27me3, reduced to lesser extent in the two clusters (Fig. 5c,d and Extended data Fig. 5e). Thus, impaired activity of PRC1 might drive de-repression of developmental genes in HD vulnerable neurons, promoting de-repression neurodevelopmental transcription factors in cluster 1 that control neuronal fate’.

We performed the analysis suggested by the reviewer, generating cluster 1’ and cluster 2’ (highly and moderately enriched in H3K27me3, respectively) using H3K27me3 data. We found that H3K27me3 was similarly reduced in R6/1 vs WT between clusters 1’ and 2’, while H2AK119ub depletion was greater for cluster 1’ vs 2’ (Fig. S5e,f)

Moreover, we generated new H3K27me3 and H2AK119ub CUT&Tag data on NeuN+ samples from the striatum of R6/1 and WT mice (3 biological replicates in each group, Fig. S5g,h,i). Above results were confirmed, since H2AK119ub reduction was greater in cluster 1 vs cluster 2, in contrast to H3K27me3, which was similarly decreased in the two clusters. See p.6:

“H2AK119ub was specifically reduced at cluster 1 promoters in R6/1 vs WT striatal samples (Fig. 5b). In contrast, H3K27me3 depletion in R6/1 vs WT striatal neurons was similar in both clusters (Fig. 5d). Additionally, H3K27ac was similarly increased in both clusters in R6/1 striatal neurons (Fig. 5d). Kmeans clustering of bivalent genes using H3K27me3 data, instead of H2AK119ub data, supported the results, showing specific decrease of H2AK119ub in gene cluster normally strongly repressed (Supplementary Fig. 5f). Finally, to better compare H2AK119ub and H3K27me3 changes in R6/1 striatal neurons, we generated H2AK119ub and H3K27me3 FANS-CUT&Tag data using same NeuN+ nuclei (Supplementary Fig. 5g,h). Data analysis confirmed that H2AK119b, in contrast to H3K27me3, was specifically reduced in cluster 1, in R6/1 vs WT striatal neurons (Supplementary Fig. 5i).”

Page 6/7. ‘Principal component analysis (PCA) analysis showed that H3K27ac and H3K27me3 samples clustered according to genotype and age (Extended Data Fig. 6a).’ Only very weakly.

We have toned down the sentence: ‘Principal component analysis (PCA) analysis showed that H3K27ac and H3K27me3 samples relatively clustered according to genotype and age (Extended Data Fig. 6a).’

Fig 6b. The raw data is noisy (is this a single replicate or sum/average of multiple)?

In previous Fig. 6b (and S6e), we merged WT and HD samples and gain space. Nonetheless, to improve data visualization, we now show WT and HD samples separately. We take the liberty of adding that this way of presenting shows that the data are not particularly noisy. Moreover, the different biological replicates are accessible through GEO and can be visualized using bigwig files.

Fig. 6e is underpowered for this analysis. 3 points is not robust enough to provide an epigenetic clock from which defined lengths of relative temporal shift can be robustly determined. Epigenetic clocks require massive amounts of data on which to model such trends.

We have rephrased our sentences and do not use any more 'epigenetic clock' to refer to H3K27ac and H3K27me3 changes over time at developmental genes:

P7: "In fact, cluster 1 euchromatin score could be used as a proxy to measure epigenetic age of HD striatal neurons, which was 3 month-older than chronological age in 6-month-old mice (Fig. 6e)."

P8: "Thus, euchromatinization rate of developmental transcription factors could be used as an aging index to show acceleration of epigenetic age in HD vulnerable neurons. The data also suggest specific role for PRC1-CBX proteins in the mechanism."

P9: "Furthermore, we defined H3K27ac/H3K27me3-based euchromatin score, which applied to subcluster of bivalent developmental transcription factors can be used as an aging index."

P10: "Here we show that cell-type-specific histone modifications (i.e. H3K27ac and H3K27me3) can also be used as a proxy to measure epigenetic age."

P10: "Future studies might determine whether the approach could be used to build histone modification-based epigenetic clock."

Figure 6K. All modules as displayed have a difference compared to WT not just 2 and 4.

We focused on modules M2 and M4, since they show differences between HD and WT (source data file).

P8: "Whereas the 4 modules were age-dependent, M2 and M4 modules also appeared related to genotype (Fig. 6k)."

Page 8. 'This suggests that epigenetic mechanism triggers early stress response in HD striatal neurons'. There is no evidence for this. Gene expression has changed so, so has the chromatin landscape. Can't tell cause and effect.

We further toned down the sentence: 'This suggests that epigenetic mechanism might trigger early stress response in HD striatal neurons'.

Minor.

'Euchromatinized' is not a commonly used word for the observed change in chromatin landscape, this should be rephrased.

The sentences p5 were rephrased: "Interestingly, we found that the top region showing increased chromatin relaxation in R6/1 vs WT striatal neurons, with most significant adjusted p-values for both depletion in H3K27me3 and enrichment in H3K27ac, was an enhancer located in the genomic region that contains PRC1 *Cbx2/4/8* paralogous genes, between *Cbx4* and *Cbx8* (Fig. 4a). Additional significantly de-repressed region was an enhancer located upstream of *Cbx4* promoter (Fig. 4a)".

The sentence p11 was rephrased: "Finally, we identified module enriched in stress response genes showing increased euchromatin score at prodromal disease stage ..."

Fig S1C – scale profiles to the same y per IP so that true comparisons of signal and differential signal can be determined.

Metaprofiles in Fig. S1c were generated using same scale.

Page 7. 'stronger contribution of H3K27me3 than H3K27ac to epigenetic aging in HD striatal neurons (Fig. 6f)' – This is correlative.

We made full sentence cautious "Our data suggest stronger contribution of H3K27me3 than H3K27ac to epigenetic aging in HD striatal neurons (Fig. 6f)". Nonetheless, the text was modified according to results of analysis suggested by reviewer 2:

P7: "Age-dependent variations of H3K27me3 and H3K27ac in cluster 1 genes suggest that H3K27me3 contributes to epigenetic age in both HD and WT striatal neurons, whereas H3K27ac may have specific

role in HD (Fig. 6f). Differential analysis of H3K27me3 and H3K27ac levels in WT samples showed that genes depleted in H3K27me3 and increased in H3K27ac with age, both displayed developmental signatures, including signature related to “Nervous system development” (Supplementary Fig. 6f). Moreover, H3K27me3-depleted genes in HD vs WT and in old vs young samples significantly overlapped (Supplementary Fig. 6g). This was also the case when considering H3K27ac-increased genes (Supplementary Fig. 6g). This indicates that common mechanism contributes to de-repression of developmental genes in normal aging and in HD. Interestingly however, the analysis also suggests greater involvement of H3K27ac in HD and of H3K27me3 in normal aging, since changes in H3K27ac and in H3K27me are predominant in HD and normal aging, respectively (Supplementary Fig. 6g).”

Fig 7C and D. Y axis values are too small to discern
This was improved.

I did not have access to Sup Tables.
Supplemental table is included.